# EQUIVARIANT NEURAL NETWORKS FOR GENERAL LINEAR SYMMETRIES ON LIE ALGEBRAS

## ABSTRACT

Encoding symmetries is a powerful inductive bias for improving the generalization of deep neural networks. However, most existing equivariant models are limited to simple symmetries like rotations, failing to address the broader class of general linear transformations, $\mathrm{GL}(n)$, that appear in many scientific domains. We introduce **Reductive Lie Neurons (ReLNs)**, a novel neural network architecture exactly equivariant to these general linear symmetries. ReLNs are designed to operate directly on a wide range of structured inputs, including general $n$-by-$n$ matrices. ReLNs introduce a novel adjoint-invariant bilinear layer to achieve stable equivariance for both Lie-algebraic features and matrix-valued inputs, *without requiring redesign for each subgroup*. This architecture overcomes the limitations of prior equivariant networks that only apply to compact groups or simple vector data. We validate ReLNs' versatility across a spectrum of tasks: they outperform existing methods on algebraic benchmarks with $\mathfrak{sl}(3)$ and $\mathfrak{sp}(4)$ symmetries and achieve competitive results on a Lorentz-equivariant particle physics task. In 3D drone state estimation with geometric uncertinaty, ReLNs jointly process velocities and covariances, yielding significant improvements in trajectory accuracy. ReLNs provide a practical and general framework for learning with broad linear group symmetries on Lie algebras and matrix-valued data.

## 1 INTRODUCTION

Leveraging the symmetries present in real-world data improves the generalization of neural networks. When a model is equivariant, i.e., its outputs transform predictably with inputs under certain group actions, it can learn more efficiently and generalize across scenarios with less data.

Most advances in equivariant deep learning target compact symmetry groups, such as rotations, $\mathrm{SO}(3)$, or isometries, $\mathrm{E}(n)$, where strong geometric and algebraic tools support robust network design (Bronstein et al., 2021; Cohen & Welling, 2016; Thomas et al., 2018; Satorras et al., 2021; Finzi et al., 2020; Geiger & Smidt, 2022). These methods have led to practical gains. However, many real-world phenomena, such as affine distortions, sensor anisotropy, and general changes of basis, are governed by the broader general linear group $\mathrm{GL}(n) = \{A \in \mathbb{R}^{n \times n} : \det(A) \neq 0\}$, for which existing methods fail to provide exact equivariance. These more general symmetries are crucial in a wide array of scientific domains, including robotics, particle physics, and computer vision, a landscape we survey in Table 1.

Naive solutions, such as flattening matrices into vectors, discard essential geometric structure, while methods based on spectral decompositions are brittle and numerically unstable (Magnus, 1985). This leaves a gap for a numerically stable neural architecture capable of processing diverse geometric data types equivariantly under arbitrary invertible linear transformations (Bronstein et al., 2021).

This work introduces Reductive Lie Neurons (ReLNs), a unified neural architecture that provides exact equivariance to the adjoint action of the general linear group, $\mathrm{GL}(n)$, and its reductive subgroups, enabling robust learning on both Lie-algebraic features and structured geometric data like covariances. We resolve the central technical obstacle for non-semisimple algebras—the degeneracy of the canonical Killing form—by introducing a learnable, provably non-degenerate, and $\mathrm{Ad}$-invariant bilinear form. This provides a mathematically stable mechanism for defining symmetry-preserving nonlinearities, pooling, and invariant layers applicable to the reductive Lie algebra $\mathfrak{gl}(n) = \mathbb{R}^{n \times n}$.

Figure 1: Examples of Lie groups and related manifolds in scientific applications. From left: the special linear group $\mathrm{SL}(3)$ (image homography), the Lorentz group $\mathrm{SO}(1,3)$ (spacetime symmetry), symplectic groups $\mathrm{Sp}(n)$ (Hamiltonian mechanics), the $\mathrm{SPD}(3) \oplus \mathbb{R}^3$ state space (probabilistic estimation), and the general linear group $\mathrm{GL}(3)$ (modeling stress-strain in continuum mechanics). See Table 1 for a more detailed survey.

Table 1: A survey of common Lie groups and their applications in equivariant deep learning.

| Group / Data Structure | Application | Reference |
|---|---|---|
| $\mathrm{SO}(3)$ | 3D Point Clouds, State Estimation | Lin et al. (2024a); Deng et al. (2021); Son et al. (2024) |
| $\mathrm{SO}^+(1,3)$ | Particle Physics, Jet Tagging | Bogatskiy et al. (2020); Finzi et al. (2021); Batatia et al. (2023) |
| $\mathrm{SU}(3)$ | Quantum Chromodynamics (QCD) | Favoni et al. (2022) |
| $\mathrm{SL}(3)$ | Homography Classification, 3D Vision | Lin et al. (2024a); Finzi et al. (2021) |
| $\mathrm{Sp}(4, \mathbb{R})$ | Hamiltonian dynamics | Lin et al. (2024a); Finzi et al. (2021) |
| $\mathrm{SPD}(n)^\dagger$ | Geometric Uncertainty Processing | - |
| $\mathrm{GL}(n)$ | General Linear Transformations | Basu et al. (2025); Finzi et al. (2021) |

[†]Not a group. $\mathrm{SPD}(n)$ is the manifold of symmetric positive-definite matrices, representable as the quotient space $\mathrm{GL}(n)/\mathrm{O}(n)$.

Our main contributions are:

1. We propose Reductive Lie Neurons (ReLNs), a novel, general-purpose, and numerically stable network architecture for exact $\mathbf{GL(n)}$ adjoint equivariance.

2. We establish a connection between classical left-action equivariance and our adjoint-action framework for orthogonal groups. Through a provably-equivariant embedding map, we show that problems defined on standard vector actions—such as Lorentz transformations in particle physics or 3D point cloud processing—can be solved within our unified architecture, obviating the need for specialized model designs.

3. We establish the framework for geometric uncertainty-aware equivariant learning, enabling models to treat matrix-valued data that transforms under congruence (e.g., covariance tensors) as geometric objects.

4. We demonstrate the effectiveness of ReLNs through extensive experiments, showing that they outperform prior methods on Lie-algebraic benchmarks and achieve significant improvements in accuracy and robustness on a challenging 3D drone state estimation task.

## 2 RELATED WORK

Encoding symmetry into neural architectures is a powerful inductive bias that improves data efficiency and generalization (Bronstein et al., 2021). In geometric deep learning, the most mature area of research focuses on enforcing equivariance to Euclidean isometries, transformations such as rotations and rigid motions formalized by Lie groups like $\mathrm{SO}(n)$ and $\mathrm{SE}(n)$. For grid-like data, foundational works include Group-Equivariant CNNs (Cohen & Welling, 2016) and Steerable CNNs (Weiler et al., 2018; Weiler & Cesa, 2021). For unstructured data, a dominant paradigm uses features associated with irreducible group representations (tensorial methods, E(n)-GNNs, and transformers (Thomas et al., 2018; Fuchs et al., 2020; Satorras et al., 2021; Batatia et al., 2022; Battiloro et al., 2025; Liao & Smidt, 2023; Assaad et al., 2023; Hutchinson et al., 2021)), with lightweight vector-based alternatives also available (Deng et al., 2021; Son et al., 2024). Theoretical analyses have also explored the universal approximation capabilities of such invariant networks (Maron et al., 2019). Complementing these specialized layers are general, model-agnostic strategies like frame averaging and canonicalization (Puny et al., 2022; Lin et al., 2024b; Kaba et al., 2023; Panigrahi & Mondal,

2024). Despite their success, most practical implementations are centered on scalar and vector features. While frameworks such as Tensor Field Networks (Thomas et al., 2018) or EMLP (Finzi et al., 2021) can handle higher-order tensor representations, scalable architectural design for general matrix-valued quantities (e.g., covariances or inertia tensors transforming as $\Sigma \mapsto R\Sigma R^\top$) is still limited in practice.

A major frontier is extending equivariance to non-compact groups like the general linear group $\mathrm{GL}(n)$ and the Lorentz group $\mathrm{SO}^+(1,3)$. Much of the work on the Lorentz group, which is critical in particle physics, has focused on designing specialized networks for the standard left action on vectors (Bogatskiy et al., 2020; Finzi et al., 2021; Batatia et al., 2023; Zhdanov et al., 2024). In contrast, we demonstrate that this left-action problem can be addressed within our universal, adjoint-equivariant framework via an embedding map.

One line of work generalizes group convolution and kernel design, often leveraging Fourier analysis (Xu et al., 2022; Helwig et al., 2023). This includes general frameworks for constructing equivariant networks on arbitrary matrix groups (Basu et al., 2025), through matrix functions (Batatia et al., 2024), and on reductive Lie groups (Batatia et al., 2023). Other approaches define kernels in the Lie algebra (Finzi et al., 2020), use Lie group decompositions for integration (Mironenco & Forré, 2024), or adapt canonicalization using infinitesimal generators of the Lie algebra (Shumaylov et al., 2025).

For these non-compact groups in general, the theoretical machinery underpinning many equivariant models does not readily apply. One line of work generalizes group convolution and kernel design, leveraging tools from Fourier analysis (Xu et al., 2022; Helwig et al., 2023), matrix functions (Batatia et al., 2024), or operating directly on reductive Lie groups (Batatia et al., 2023). Other methods focus on the Lie algebra, either by defining kernels within the algebra itself (Finzi et al., 2020) or using Lie group decompositions for integration (Mironenco & Forré, 2024; Shumaylov et al., 2025). Another approach leverages differential geometry, using tools such as partial differential operators (He et al., 2022; Shen et al., 2020; Jenner & Weiler, 2022) and the algebra of differential invariants (Sangalli et al., 2022; Li et al., 2024) to construct equivariant layers. A different generalist approach, taken by methods like the Equivariant MLP and G-RepsNet (Finzi et al., 2021; Basu et al., 2025), is to solve the equivariance constraint algebraically, though this often lacks specialized inductive biases like locality.

In contrast to these manifold-focused approaches, a parallel line of work operates directly on the Lie algebra. Lin et al. (2024a) introduced Lie Neurons, establishing a framework for adjoint-equivariance, but their method is restricted to semisimple Lie algebras where the Killing form is non-degenerate. This limitation precludes direct application to the **reductive** but non-semisimple algebra $\mathfrak{gl}(n)$, whose degenerate Killing form poses a central challenge. The high computational cost of group convolution methods, combined with the semisimple constraints of existing Lie-algebraic techniques, limits their direct applicability to robotics tasks that involve real-time processing, noisy measurements, or uncertainty estimation (Eschmann et al., 2024; Yu & Sun, 2024). While traditional, model-based algorithms like Kalman filters are designed to respect this geometry (Barrau & Bonnabel, 2016; Hartley et al., 2020), they lack the flexibility.

Our work confronts this challenge with ReLNs, a practical, numerically well-conditioned architecture for the full reductive group $\mathrm{GL}(n)$ and its subgroups. By introducing a learnable, non-degenerate, and $\mathrm{Ad}$-invariant bilinear form, our framework overcomes the key obstacle of the degenerate Killing form in $\mathfrak{gl}(n)$. This unified algebraic approach extends beyond strict Lie algebra elements to other geometric features, such as covariance matrices, that transform under congruence, providing a practical tool that sidesteps the complexities of prior methods requiring group integration or degenerate invariants. Figure 2 situates ReLNs as a general framework for $\mathrm{GL}(n)$ transformations, distinguishing our approach from architectures tailored to specialized subgroups.

## 3 PRELIMINARIES

Our work builds equivariant networks on the Lie algebra $\mathfrak{g} = \mathrm{Lie}(G)$, the tangent space of a Lie group $G$ at the identity $e$. We focus on the general linear group $\mathrm{GL}(n)$ and its Lie algebra $\mathfrak{gl}(n)$, aiming for equivariance under the adjoint action $\mathrm{Ad}_g : X \mapsto gXg^{-1}$, $g \in \mathrm{GL}(n)$, $X \in \mathfrak{gl}(n)$. A central challenge arises from the structure of $\mathfrak{gl}(n)$: it is a **reductive** Lie algebra, and the canonical invariant inner product, the Killing form, is degenerate on $\mathfrak{gl}(n)$. This degeneracy poses a critical

Figure 2: A taxonomy of selected representative equivariant neural architectures, categorized by the symmetries to which they are equivariant. This diagram situates our work, ReLNs, among other notable methods that are often specialized for subgroups such as $SL(n)$, $SO(n)$, or the Euclidean group $E(n)$. An asterisk ($*$) denotes methods equivariant to the group's adjoint action.

problem, as it prevents the construction of expressive nonlinear layers, causing them to collapse into linear maps and severely limiting the model's expressive power. We address this by introducing a learnable, non-degenerate bilinear form, enabling fully nonlinear equivariant operations. For details of Lie theory and background, see Appendix A.

# 4 REDUCTIVE LIE NEURONS: ARCHITECTURE

We present ReLNs, a framework for building deep networks equivariant to the adjoint action of the general linear group $GL(n)$. The design centers on a learnable, non-degenerate, $\mathrm{Ad}$-invariant bilinear form on the reductive Lie algebra $\mathfrak{gl}(n)$ and a complete toolbox of equivariant linear maps, nonlinearities, pooling, and invariant readouts.

## 4.1 A GENERAL $\mathrm{Ad}$-INVARIANT BILINEAR FORM FOR REDUCTIVE LIE ALGEBRAS

The primary obstacle to applying Lie-algebraic methods such as Lie Neurons (Lin et al., 2024a) to $\mathfrak{gl}(n)$ is the degeneracy of its Killing form. We resolve this by constructing a modified bilinear form $\widetilde{B}$ that restores non-degeneracy while preserving the crucial $\mathrm{Ad}$-invariance property.

**Definition 4.1** (Modified Bilinear Form on a Reductive Lie Algebra). If $\mathfrak{g}$ is reductive, then $\mathfrak{g} = \mathfrak{z}(\mathfrak{g}) \oplus [\mathfrak{g}, \mathfrak{g}]$, where $\mathfrak{z}(\mathfrak{g})$ is the center. Choose any $\mathrm{Ad}$-invariant inner product $\langle \cdot, \cdot \rangle_{\mathfrak{z}}$ on $\mathfrak{z}(\mathfrak{g})$ (for connected $G$ this is automatic since $\mathrm{Ad}|_{\mathfrak{z}(\mathfrak{g})} : G \to GL(\mathfrak{z}(\mathfrak{g}))$ is locally constant. See Appendix A.2 for the formal definition of $\mathrm{Ad}$) . For $Z_i \in \mathfrak{z}(\mathfrak{g})$ and $X_i \in [\mathfrak{g}, \mathfrak{g}]$ define

$$\widetilde{B}(Z_1+X_1,\, Z_2+X_2) := \langle Z_1, Z_2 \rangle_{\mathfrak{z}} + B(X_1, X_2), \tag{1}$$

where $B$ denotes the Killing form on $[\mathfrak{g}, \mathfrak{g}]$.

**Proposition 4.1.** The bilinear form $\widetilde{B}$ is symmetric, $\mathrm{Ad}$-invariant, and nondegenerate. Moreover, $\mathfrak{z}(\mathfrak{g})$ and $[\mathfrak{g}, \mathfrak{g}]$ are $\widetilde{B}$-orthogonal, with $\widetilde{B}|_{[\mathfrak{g}, \mathfrak{g}]} = B$ and $\widetilde{B}|_{\mathfrak{z}(\mathfrak{g})} = \langle \cdot, \cdot \rangle_{\mathfrak{z}}$.

*Proof sketch.* $B$ vanishes on $\mathfrak{z}(\mathfrak{g})$ and is $\mathrm{Ad}$-invariant; by construction $\langle \cdot, \cdot \rangle_{\mathfrak{z}}$ is $\mathrm{Ad}$-invariant. Symmetry is immediate. Nondegeneracy follows since $B$ is nondegenerate on the semisimple ideal and $\langle \cdot, \cdot \rangle_{\mathfrak{z}}$ is nondegenerate on the center; orthogonality holds because $B(\mathfrak{z}(\mathfrak{g}), [\mathfrak{g}, \mathfrak{g}]) = 0$. $\square$

For our primary case $\mathfrak{g} = \mathfrak{gl}(n, \mathbb{R}) = \mathbb{R}I \oplus \mathfrak{sl}(n, \mathbb{R})$ we choose the canonical trace-based form

$$\widetilde{B}(X, Y) = 2n \cdot \mathrm{tr}(XY) - \mathrm{tr}(X)\,\mathrm{tr}(Y), \tag{2}$$

This form $\widetilde{B}$ is the fundamental tool used throughout the ReLN architecture.

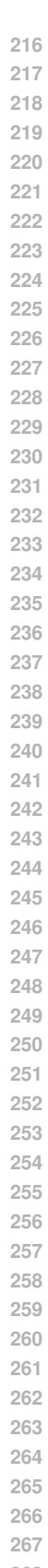

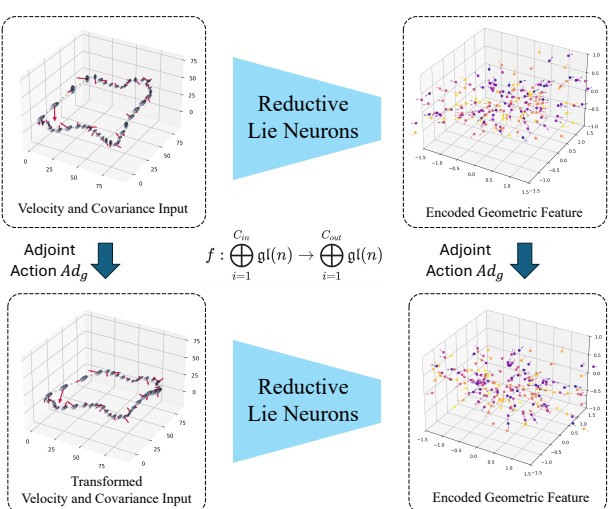

Figure 3: Adjoint equivariance using a unified representation for diverse geometric inputs. Our framework embeds inputs with different transformation rules, such as velocity ($v \mapsto Rv$) and covariance ($\Sigma \mapsto R\Sigma R^T$), into a common Lie algebra. Therefore, they transform under the same adjoint action $\mathrm{Ad}_g$, with which our network $f$ commutes as shown in the diagram.

**Verification and Relation to Prior Bilinear Forms.** Our concrete form in Eq. 2 satisfies the conditions of Proposition 4.1. Decomposing a matrix $X = X_0 + \frac{1}{n}\mathrm{tr}(X)I$ (where $X_0 \in \mathfrak{sl}(n)$) reveals that our form separates orthogonally:

$$\widetilde{B}(X,Y) = \underbrace{2n \cdot \mathrm{tr}(X_0 Y_0)}_{B_{\mathfrak{sl}(n)}(X_0,Y_0)} + \underbrace{\mathrm{tr}(X)\mathrm{tr}(Y)}_{\text{Inner product on } \mathbb{R}I} . \tag{3}$$

This decomposition directly shows how $\widetilde{B}$ serves as a generalization of prior work. The first term is the Killing form on the semisimple part, which is the tool used in Lie Neurons (Lin et al., 2024a). The second term is a standard inner product on the center, which, under the isomorphism $\mathfrak{so}(3) \simeq \mathbb{R}^3$, recovers the dot product used in Vector Neurons (Deng et al., 2021). Our single form thus unifies these approaches, extending to the full reductive algebra $\mathfrak{gl}(n)$ (details can be found in Appendix C).

## 4.2 THE RELN LAYER TOOLBOX

We represent multi-channel input as $x \in \mathbb{R}^{K \times C}$, where each column $x_c \in \mathbb{R}^K$ corresponds to a matrix $X_c \in \mathfrak{g}$ (via the vee/hat isomorphism, Appendix A) .

**ReLN-Linear.** A linear map applied to the channel dimension $f(x; W) = xW$ with $W \in \mathbb{R}^{C \times C'}$ is strictly equivariant: the group acts on the left (geometric dimension) while $W$ acts on the right (channel dimension), and thus these operations commute (formal proof in Appendix D).

**Equivariant Nonlinearities.** We introduce two complementary nonlinear primitives; full definitions, parameterizations, and stability prescriptions are deferred to Appendix D.

- **ReLN-ReLU**: To overcome the non-equivariance of elementwise activations, we use our form $\widetilde{B}$ to define a directional nonlinearity. Each vector feature $x_c$ is rectified along a learnable direction $d_c$ via update rule:

- **ReLN-ReLU** Using $\widetilde{B}$ to build an invariant gate, each channel feature is updated by

$$x'_c = x_c + \max\left(0, \ \widetilde{B}(x_c^\wedge, d_c^\wedge)\right) d_c, \tag{4}$$

where $x_c^\wedge, d_c^\wedge \in \mathfrak{g}$ are the matrix forms of the channel feature and a learnable direction. Because $\widetilde{B}(\cdot, \cdot)$ is $\mathrm{Ad}$-invariant, the scalar gate is invariant and the vector update is equivariant.

Table 2: Platonic solid classification (mean $\pm$ std over 5 runs). ID = in-distribution; RC = rotated-camera (10 random $\mathrm{SO}(3)$ test rotations). "+Aug" denotes training with random $\mathrm{SO}(3)$ augmentation of input homographies. Higher is better ($\uparrow$).

| Model | # Params | ID Acc (mean $\pm$ std) | RC Acc (mean $\pm$ std) |
|---|---|---|---|
| MLP | 206,339 | 95.76% $\pm$ 0.65% | 36.54% $\pm$ 0.99% |
| MLP + Aug | 206,339 | 81.47% $\pm$ 0.77% | 81.20% $\pm$ 2.34% |
| MLP (wider) | 411,479 | 96.82% $\pm$ 0.53% | 36.55% $\pm$ 0.34% |
| MLP (wider) + Aug | 411,479 | 85.22% $\pm$ 1.46% | 83.43% $\pm$ 0.51% |
| Lie Neurons | 331,272 | 99.62% $\pm$ 0.25% | 99.61% $\pm$ 0.14% |
| ReLN (Ours) | 331,272 | **99.78% $\pm$ 0.04%** | **99.78% $\pm$ 0.04%** |

- **ReLN-Bracket.** Following prior work (Lin et al., 2024a), we include a layer that leverages the Lie bracket (matrix commutator). This operation is an $\mathrm{Ad}$-equivariant primitive on the Lie algebra that creates nonlinear interactions by measuring the non-commutativity of features. The layer applies two independent linear maps, parameterized by weights $W_a, W_b \in \mathbb{R}^{C \times C}$, to the input channels $x_{\mathrm{in}}$ to produce two intermediate features, computes their commutator, and injects the vectorized result as a shared residual:

$$x_{\mathrm{out}} = x_{\mathrm{in}} + \left( \left[ (x_{\mathrm{in}} W_a)^\wedge, (x_{\mathrm{in}} W_b)^\wedge \right] \right)^\vee. \tag{5}$$

**Equivariant Pooling and Invariant Layers.** The final components of the ReLN toolbox enable feature aggregation and the production of invariant outputs.

- **Max-Killing Pooling:** To aggregate a set of features $\{X_n\}_{n=1}^N$, where each $X_n$ is a multi-channel feature tensor, this layer selects the representative feature with the maximal projection onto a learnable direction. For each channel $c$, the index is found via $n^*(c) = \arg\max_n \widetilde{B}(X_{n,c}, D_{n,c})$, and the pooled feature is $X_c^{\max} = X_{n^*(c),c}$.
- **Invariant Layer:** To produce a group-invariant output, this layer contracts feature $X_c$ using the form $\widetilde{B}$. The resulting scalar, $y_c = \widetilde{B}(X_c, X_c)$, is invariant by construction.

**Unifying geometric representations.** By operating directly on $n \times n$ matrix representations, ReLNs provide a unified primitive for vectors, matrices, and higher-order geometric objects (e.g., covariances). This allows ReLNs to handle a broader class of geometric inputs without resorting to separate specialized architectures; empirical validation is presented in Section 5.3.

## 5 EXPERIMENTS

We evaluate ReLNs on a suite of tasks designed to highlight two complementary strengths: *algebraic generality* on benchmarks and *practical efficacy* on a challenging, uncertainty-aware robotics task. We compare against standard non-equivariant baselines (MLP, ResNet), the original Lie Neurons, and a Vector Neurons–style baseline adapted for covariance inputs by eigendecomposition. For the Top-Tagging benchmark we also report results versus established physics models used in prior work.

### 5.1 ALGEBRAIC BENCHMARKS ON SEMISIMPLE LIE ALGEBRAS

To verify that our general $\mathfrak{gl}(n)$ framework correctly generalizes to semisimple subalgebras, we evaluate ReLN on two Lie-algebraic benchmarks first introduced by Lie Neurons (Lin et al., 2024a).

#### 5.1.1 PLATONIC SOLID CLASSIFICATION

We first validate our model on the Platonic solid classification benchmark from (Lin et al., 2024a), testing the adjoint-equivariance where camera rotations induce a conjugation action on inter-face homographies in $\mathrm{SL}(3)$. Full experimental details are provided in Appendix E.2.

The results, summarized in Table 2, confirm that non-equivariant baselines fail to generalize to rotated camera views. This fundamental weakness is not resolved by data augmentation or increased model capacity, as our wider MLP variant with approximately double the parameters shows negligible

Table 3: Regression performance and invariance error on $\mathfrak{sp}(4)$. "Training Aug." indicates whether $\mathrm{Sp}(4)$ conjugation was applied during training.

| Model | Training Aug. | # Params | Test Aug. | | Invariance Error |
|-------|---------------|----------|-----------|-----|------------------|
| | | | ID | SP(4) | |
| MLP 256 | Id | 137,217 | 0.126 | 1.360 | 0.722 |
| | SP(4) | 137,217 | 0.192 | 0.587 | 0.476 |
| MLP 512 | Id | 536,577 | 0.107 | 0.906 | 0.585 |
| | SP(4) | 536,577 | 0.123 | 0.446 | 0.374 |
| Lie Neurons | Id | 263,170 | $5.83 \times 10^{-4}$ | $5.84 \times 10^{-4}$ | $3.84 \times 10^{-7}$ |
| ReLN (ours) | Id | 263,170 | $5.14 \times 10^{-4}$ | $5.14 \times 10^{-4}$ | $4.73 \times 10^{-7}$ |

improvement on the out-of-distribution test set. In contrast, the ReLN model achieves near-perfect accuracy with robustness, matching the performance of the Lie Neurons while demonstrating improved results. Importantly, this result validates that our general $\mathfrak{gl}(n)$ framework operates effectively on common semisimple subalgebras, as its built-in adjoint-equivariance on the parent group yields robust behavior when restricted to subgroups like $\mathrm{SO}(3)$ and $\mathrm{SL}(3)$.

### 5.1.2 INVARIANT FUNCTION REGRESSION ON $\mathfrak{sp}(4)$.

To further probe our framework's algebraic generality, our second benchmark involves regressing a highly nonlinear invariant function on the real symplectic Lie algebra $\mathfrak{sp}(4, \mathbb{R})$. The symplectic algebra $\mathfrak{sp}(2n)$ is the mathematical foundation of Hamiltonian mechanics, which describes any physical system where energy is conserved. Our target is a scalar invariant defined for pairs $X, Y \in \mathfrak{sp}(4, \mathbb{R})$:

$$g(X, Y) = \sin\big(\operatorname{Tr}(XY)\big) + \cos\big(\operatorname{Tr}(YY)\big) - \tfrac{1}{2}\operatorname{Tr}(YY)^3 + \det(XY) + \exp\big(\operatorname{Tr}(XX)\big). \quad (6)$$

We generate a dataset of 10k training and 10k test pairs by sampling from $\mathfrak{sp}(4, \mathbb{R})$. We compare ReLN against MLP baselines (trained with and without $\mathrm{Sp}(4)$ data augmentation) and the original Lie Neurons. At test time, we report the standard MSE, MSE averaged over 500 random adjoint actions, and the invariance error.

As shown in Table 3, non-equivariant MLPs are orders of magnitude less accurate and exhibit high invariance error, failing to learn the group structure even with data augmentation. Our ReLN model not only achieves the lowest MSE and near-zero invariance error, but also shows a modest but consistent improvement over Lie Neurons. This suggests that our non-degenerate bilinear form provides not only theoretical generality but also superior numerical conditioning in practice.

### 5.2 PARTICLE PHYSICS WITH LORENTZ GROUP $\mathrm{SO}(1, 3)$ EQUIVARIANCE

We test our framework on the Top-Tagging benchmark (Kasieczka et al., 2019), a task to distinguish particle jets originating from top quarks against a large background from Quantum Chromodynamics. Because these relativistic collisions are subject to the symmetries of spacetime, the task requires equivariance under the Lorentz group $\mathrm{SO}(1, 3)$. We solve this left-action equivariant learning problem by introducing a map that embeds the four-momentum $p \in \mathbb{R}^{1,3}$ of each constituent particle, which combines its energy and 3D momentum, into the Lie algebra $\mathfrak{gl}(5)$ as $\varphi(p) = \begin{pmatrix} 0_{4\times4} & p \\ p^\top \eta & 0 \end{pmatrix}$, where $\eta = \operatorname{diag}(-1, 1, 1, 1)$ is the Minkowski metric. As proven in Appendix F.1, this embedding unifies left- and adjoint-action equivariance within a single Lie-algebraic framework. We adapt the LorentzNet architecture by replacing its invariant feature computation with our proposed bilinear form. To ensure a fair comparison, we created a parameter-matched version of LorentzNet. As shown in Table 4, our model achieves competitive overall performance while demonstrating an advantage on the background rejection metric. This result shows that our general Lie-algebraic approach can effectively unify adjoint- and left-action equivariance in a parameter-efficient manner.

### 5.3 DRONE STATE ESTIMATION WITH GEOMETRIC UNCERTAINTY

We test our framework on a challenging drone state estimation task using a large-scale dataset of aggressive, highly dynamic flights. The objective is to recover a 3D trajectory from a sequence

Table 4: Comparison of performance on the Top-Tagging dataset. Rej@30% denotes the background rejection at 30% signal efficiency (higher is better). Benchmark scores are as reported in the original publications.

| Architecture | #Params | Accuracy | AUC | Rej@30% | Reference |
|---|---|---|---|---|---|
| PELICAN | 45k | 0.943 | 0.987 | $2289 \pm 204$ | Bogatskiy et al. (2022) |
| LorentzNet (original) | 224k | 0.942 | 0.987 | $2195 \pm 173$ | Gong et al. (2022) |
| LorentzNet (param-matched) | 84k | 0.942 | 0.987 | $1821 \pm 94$ | Our reproduction |
| LGN | 4.5k | 0.929 | 0.964 | $435 \pm 95$ | Bogatskiy et al. (2020) |
| BIP | 4k | 0.931 | 0.981 | $853 \pm 68$ | Munoz et al. (2022) |
| partT | 2.14M | 0.940 | 0.986 | $1602 \pm 81$ | Qu et al. (2022) |
| ParticleNet | 498k | 0.938 | 0.985 | $1298 \pm 46$ | Qu & Gouskos (2020) |
| EFN | 82k | 0.927 | 0.979 | $633 \pm 31$ | Komiske et al. (2019) |
| TopoDNN | 59k | 0.916 | 0.972 | $295 \pm 5$ | Pearkes et al. (2017) |
| LorentzMACE | 228k | 0.942 | 0.987 | $1935 \pm 85$ | Batatia et al. (2023) |
| **ReLN (Ours)** | 84k | 0.942 | 0.987 | $1979 \pm 87$ | |

of noisy velocity measurements and their corresponding time-varying covariances, where each covariance matrix quantifies the uncertainty of its associated velocity measurement. This setup tests a model's ability to jointly process vector ($\mathbf{v}$) and matrix ($C$) data in a geometrically consistent and uncertainty-aware manner.

**Experimental Setup.** We created a large-scale synthetic dataset of 200 aggressive drone trajectories, over 13 hours of challenging, high-speed flight (details in Appendix G). The network is trained to regress the 3D position from a sequence of velocity and covariance measurements within a time window. Our model is compared against non-equivariant ResNets and an $\mathrm{SO}(3)$-equivariant baseline using Vector Neurons (VN). Since VNs cannot directly process matrix inputs, we implement a method that handles covariance matrices via an eigendecomposition-based strategy (details in Appendix G.3).

In contrast, our ReLN architecture treats both velocity and covariance as unified geometric objects within a single algebraic space, $\mathfrak{gl}(3)$. Specifically, the velocity vector $\mathbf{v} \in \mathbb{R}^3$ is lifted to its matrix representation in $\mathfrak{so}(3) \subset \mathfrak{gl}(3)$. For the time-varying covariance $C$, we explore two representations that also reside in $\mathfrak{gl}(3)$: (1) the matrix $C$ directly, and (2) its matrix logarithm $\log C$, which respects the geometry of the $\mathrm{SPD}(3)$ manifold. This unified representation ensures a measurement and its uncertainty transform consistently under the adjoint action of $\mathrm{SO}(3)$. The final velocity estimate is then equivariantly extracted by projecting the network's matrix output onto its skew-symmetric component. We test three ReLN variants: (1) velocity only, (2) velocity + covariance, and (3) velocity + log-covariance. Full implementation details are provided in Appendix G.

### 5.3.1 RESULTS AND ANALYSIS

Table 5: Performance on the drone trajectory dataset[†]. Best result in each column is shown in **bold**.

| Model | ID | | | SO(3) | | |
|---|---|---|---|---|---|---|
| | ATE | ATE$_\%$ | RPE | ATE | ATE$_\%$ | RPE |
| *Non-Equivariant Baselines* | | | | | | |
| ResNet (Velocity only) | 208.07 | 95.06 | 107.60 | 217.02 | 100.39 | 111.29 |
| ResNet (Velocity + Covariance) | 205.11 | 94.94 | 106.07 | 213.26 | 98.90 | 109.37 |
| *Equivariant Baselines* | | | | | | |
| VN (Velocity only) | 17.36 | 7.52 | 13.51 | 17.36 | 7.52 | 13.51 |
| VN (Velocity + Covariance) | 191.78 | 88.66 | 98.39 | 190.22 | 88.47 | 98.26 |
| *Our Equivariant Models* | | | | | | |
| ReLN (Velocity only) | 16.85 | 7.31 | 12.7 | 16.85 | 7.31 | 12.7 |
| ReLN (Velocity + Covariance) | 16.49 | 7.21 | 13.02 | 16.49 | 7.21 | 13.02 |
| ReLN (Velocity + log-Covariance) | **13.92** | **5.99** | **11.04** | **13.92** | **5.99** | **11.04** |

[†] Abbreviations and units: **ID** = in-distribution test (no rotation), **SO(3)** = test-time random SO(3) rotations. ATE = absolute trajectory error (meters), ATE$_\%$ = ATE relative to trajectory length (%), RPE = relative pose error (meters).

Our experiments, summarized in Table 5 and Figure 4, reveal a clear performance hierarchy where the geometric representation of features is the critical factor.

**Non-equivariant models fail to generalize.** Non-equivariant models like ResNets fail to generalize to rotated trajectories, and their performance does not improve when provided with covariance data.

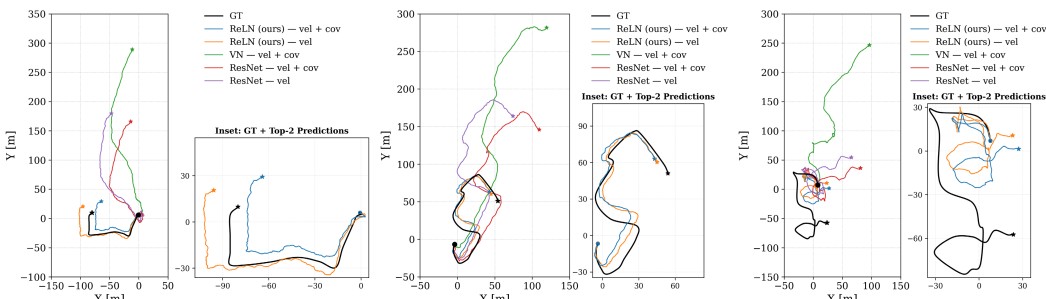

Figure 4: Qualitative comparison of the best-case (middle), average (left), and challenging (right) test sequences. ReLN models consistently track the ground truth (black) with high fidelity, especially when leveraging covariance. Insets provide a magnified view of the two best-performing variants (ours) to highlight their accuracy.

**Equivariance is necessary, but implementation design critically affects performance.** Both VNs and ReLNs establish strong velocity-only equivariant baselines. ReLNs show improved performance due to its richer feature learning from both the Lie bracket and our bilinear nonlinearity form. However, a method for incorporating covariance is crucial. The VN baseline, which decomposes covariance, degrades the performance. This result confirms that separating the principle axes and its corresponding scales (eigenvalues) prevents the network from learning the structure between a measurement and its uncertainty.

**Joint processing of velocity and covariance further improves performance.** Integrating covariance as a geometric object in our framework further reduces the ATE, showcasing the performance gains from unified geometric representation on symmetry-preserving uncertainty handling.

**Log-covariance ablation highlights mathematical and practical contributions.** Finally, the ReLN (Velocity + $\log$-Covariance) model achieves the best performance. By processing covariance matrices via the matrix logarithm, we demonstrate the power of extending Lie-algebraic architectures to operate on manifold-valued data, such as the $\mathrm{SPD}(3)$ matrices representing geometric uncertainty. This success establishes ReLNs as a practical and high-performance framework for uncertainty-aware learning in dynamical systems.

**Discussion** Our results indicate that ReLNs operate effectively as a *geometry- and uncertainty-aware estimator* that generalizes across random measurement-frame changes (i.e., arbitrary 3D rotations). While the network is not a classical recursive Markovian filter, it learns to integrate velocity measurements with their associated covariances using uncertainty-dependent weighting, producing robust and accurate trajectory estimates. This behavior suggests ReLNs are suitable as modular components in downstream systems that require handling of matrix-valued uncertainty. Future work will investigate theoretical guarantees for the learned weighting, extensions to other geometric matrix representations, and scalability to higher-dimensional structured geometric inputs.

## 6 CONCLUSION

This work introduces ReLNs, a unified neural architecture that provides exact equivariance to the adjoint action of the general $n \times n$ matrix algebra $\mathfrak{gl}(n)$ and its subgroups. ReLNs enable efficient learning on Lie-algebraic features and structured geometric data, such as covariance matrices. Furthermore, our work establishes a unified Lie-algebraic framework that handles both classical left-action symmetries on vectors and native adjoint-actions on matrices within a single architecture. ReLNs achieve state-of-the-art results on benchmarks and deliver large gains in a challenging drone state estimation task by integrating uncertainty. We'll apply our equivariant matrix processing capability to a wider array of physical systems, including the dynamics of articulated robots and large-scale sensor fusion, to further expand the boundaries of geometric deep learning.

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

# A  LIE-THEORETIC PRELIMINARIES

This appendix provides an overview of key concepts and derivations from Lie group theory relevant to our construction of $\mathrm{GL}(n)$-equivariant neural networks referenced in the main text.

## A.1  LIE GROUPS, LIE ALGEBRAS, HAT/VEE

A *matrix Lie group* $G \subset \mathrm{GL}(n)$ is a smooth subgroup of invertible matrices. Its Lie algebra $\mathfrak{g} = \mathrm{Lie}(G)$ is the tangent space at the identity and is identified with a subspace of $\mathfrak{gl}(n)$. Fix a basis $\{E_i\}_{i=1}^m$ of $\mathfrak{g}$. The coordinate maps are:

$$\wedge : \mathbb{R}^m \to \mathfrak{g}, \quad x = (x_i) \mapsto x^\wedge = \sum_i x_i E_i, \qquad \vee : \mathfrak{g} \to \mathbb{R}^m, \quad X \mapsto X^\vee. \tag{7}$$

These maps let us implement algebra-valued features as Euclidean vectors in code.

The associated Lie algebra $\mathfrak{g} = \mathrm{Lie}(G)$ is the tangent space at the identity element $e \in G$. It carries a bilinear, antisymmetric product called the *Lie bracket*, given by

$$[A, B] = AB - BA, \tag{8}$$

in the case of $\mathrm{GL}(n)$ which captures the infinitesimal structure of the group near the identity. The bracket quantifies non-commutativity of generators: $[A, B] = 0$ implies commutativity, whereas $[A, B] \neq 0$ indicates a non-trivial interaction.

## A.2  REPRESENTATIONS AND THE ADJOINT

A representation $\Phi : G \to \mathrm{GL}(V)$ differentiates to $\phi : \mathfrak{g} \to \mathfrak{gl}(V)$ by

$$\phi(X) = \left.\frac{d}{dt}\right|_{t=0} \Phi(\exp(tX)). \tag{9}$$

The adjoint representation $\mathrm{Ad} : G \to \mathrm{GL}(\mathfrak{g})$ is defined to be the differential of group conjugation at the identity

$$\mathrm{Ad}_g(X) = \left.\frac{d}{dt}\right|_{t=0} g(\exp(tX))g^{-1}. \tag{10}$$

Therefore we get a map $\mathrm{Ad} : G \to \mathrm{GL}(\mathfrak{g})$. For matrix groups, this is given

$$\mathrm{Ad}_g(X) = gXg^{-1}, \qquad g \in G, \ X \in \mathfrak{g}, \tag{11}$$

and differentiating yields the Lie-algebra adjoint $\mathrm{ad}_X(Y) = [X, Y]$. One checks

$$\mathrm{Ad}_g([X, Y]) = [\mathrm{Ad}_g X, \mathrm{Ad}_g Y], \qquad \mathrm{ad}_X([Y, Z]) = [\mathrm{ad}_X Y, Z] + [Y, \mathrm{ad}_X Z]. \tag{12}$$

## A.3  VECTORIZED ADJOINT

Using the hat/vee maps, the adjoint action on the Lie algebra induces a corresponding action on the vector coordinates. This vectorized action is a linear map represented by a matrix:

$$\mathrm{Ad}_g^m : \mathbb{R}^m \to \mathbb{R}^m, \qquad \mathrm{Ad}_g^m(x) = (\mathrm{Ad}_g(x^\wedge))^\vee. \tag{13}$$

In practice we precompute or assemble the $m \times m$ matrix representing $\mathrm{Ad}_g^m$ (or apply it implicitly) to implement left-multiplicative equivariant layers that act on vector features.

## A.4  STRUCTURE OF LIE ALGEBRA: SEMISIMPLICITY AND REDUCTIVITY

**Definition A.1** (Semisimple and Reductive Lie Algebras)**.** A Lie algebra $\mathfrak{g}$ is:

- *Semisimple* if it is a direct sum of simple Lie algebras (i.e., non-abelian and having no nontrivial ideals).

- *Reductive* if it decomposes as $\mathfrak{g} = \mathfrak{s} \oplus \mathfrak{z}$, where $\mathfrak{s}$ is semisimple and $\mathfrak{z}$ is the center (an abelian Lie subalgebra).

**Example A.1.** The Lie algebra $\mathfrak{gl}(n)$ decomposes as:

$$\mathfrak{gl}(n) = \mathfrak{sl}(n) \oplus \mathbb{R}I, \tag{14}$$

where $\mathfrak{sl}(n)$ (traceless matrices) is semisimple, and $\mathbb{R}I$ (scalar matrices) forms the center.

This decomposition highlights the non-semisimple nature of $\mathfrak{gl}(n)$, which plays a critical role in understanding the degeneracy of certain invariant forms such as the Killing form. This degeneracy hinders the application of standard tools in Lie-theoretic deep learning. Our work addresses this issue in the context of $\mathrm{GL}(n)$-equivariant architectures in lie algebra $\mathfrak{gl}(n)$.

**Theorem A.1** (Cartan criterion; standard). *The Killing form $B(X, Y) = \mathrm{tr}(\mathrm{ad}_X \circ \mathrm{ad}_Y)$ is non-degenerate iff $\mathfrak{g}$ is semisimple.*

In particular, because $\mathfrak{gl}(n)$ contains the central scalar direction $\mathbb{R}I$ with $\mathrm{ad}_I = 0$, the Killing form is degenerate on $\mathfrak{gl}(n)$.

### A.5 INVARIANT BILINEAR FORMS; TRACE FORM

**Definition A.2.** A bilinear form $B : \mathfrak{g} \times \mathfrak{g} \to \mathbb{R}$ is $\mathrm{Ad}$-*invariant* if

$$B(\mathrm{Ad}_g X, \mathrm{Ad}_g Y) = B(X, Y) \quad \forall g \in G. \tag{15}$$

On semisimple algebras the Killing form provides such an invariant, non-degenerate form. On $\mathfrak{gl}(n)$ we instead use the *trace form*:

$$\langle X, Y \rangle_{\mathrm{tr}} = \mathrm{tr}(XY). \tag{16}$$

**Proposition A.1.** The trace form is $\mathrm{Ad}$-invariant on $\mathfrak{gl}(n)$:

$$\mathrm{tr}((gXg^{-1})(gYg^{-1})) = \mathrm{tr}(gXYg^{-1}) = \mathrm{tr}(XY). \tag{17}$$

*Proof.* $\mathrm{tr}((gXg^{-1})(gYg^{-1})) = \mathrm{tr}(gXYg^{-1}) = \mathrm{tr}(XY)$ by cyclicity of trace. $\square$

The trace form is non-degenerate as a bilinear form on the vector space $\mathfrak{gl}(n)$ and therefore provides a practical substitute for the Killing form when designing $\mathrm{Ad}$-invariant bilinear layers on $\mathfrak{gl}(n)$.

### A.6 KILLING FORM ON $\mathfrak{sl}(n)$

Restricted to $\mathfrak{sl}(n)$ the Killing form simplifies and is non-degenerate; one frequently uses the proportionality $B(X, Y) \propto \mathrm{tr}(XY)$ on $\mathfrak{sl}(n)$.

## B PROOFS OF KEY THEOREMS

### B.1 PROOF OF NON-DEGENERACY AND AD-INVARIANCE OF MODIFIED KILLING FORM $B_e$

Let $\mathfrak{g}$ be a real reductive Lie algebra.

**Definition B.1** (Reductive decomposition). A Lie algebra $\mathfrak{g}$ is *reductive* if $\mathfrak{g} = \mathfrak{z}(\mathfrak{g}) \oplus [\mathfrak{g}, \mathfrak{g}]$, where $\mathfrak{z}(\mathfrak{g})$ is the center and $[\mathfrak{g}, \mathfrak{g}]$ is semisimple. This decomposition is canonical (both summands are ideals).

**Definition B.2** (Modified Killing form on a reductive Lie algebra). Fix any symmetric, positive–definite inner product $\langle \cdot, \cdot \rangle_{\mathfrak{z}}$ on $\mathfrak{z}(\mathfrak{g})$, and let $B$ denote the Killing form on the semisimple ideal $[\mathfrak{g}, \mathfrak{g}]$. For $Z_i \in \mathfrak{z}(\mathfrak{g})$ and $X_i \in [\mathfrak{g}, \mathfrak{g}]$ define

$$\widetilde{B}(Z_1 + X_1, Z_2 + X_2) := \langle Z_1, Z_2 \rangle_{\mathfrak{z}} + B(X_1, X_2). \tag{18}$$

*Remark* 1 (Canonicity). On $[\mathfrak{g}, \mathfrak{g}]$ the restriction (Killing form) is canonical. On $\mathfrak{z}(\mathfrak{g})$ there is no canonical choice; any $\mathrm{Ad}$-invariant positive-definite inner product works. The choice we make in the case of $\mathfrak{gl}(n)$ ensures that it agrees with the Killing form on the semisimple part $\mathfrak{sl}(n)$, and the center $\mathbb{R}I$ is normalized by a natural trace scale.

**Proposition B.1** (Block–orthogonality and restrictions). With notation as above,

$$\widetilde{B}\big(\mathfrak{z}(\mathfrak{g}), [\mathfrak{g}, \mathfrak{g}]\big) = 0, \qquad \widetilde{B}|_{\mathfrak{z}(\mathfrak{g})} = \langle \cdot, \cdot \rangle_{\mathfrak{z}}, \qquad \widetilde{B}|_{[\mathfrak{g}, \mathfrak{g}]} = B. \tag{19}$$

**Proposition B.2** (Non–degeneracy). $\widetilde{B}$ is nondegenerate on $\mathfrak{g}$.

*Proof.* Let $X = Z + W$ with $Z \in \mathfrak{z}(\mathfrak{g})$ and $W \in [\mathfrak{g}, \mathfrak{g}]$. If $\widetilde{B}(X, \cdot) \equiv 0$, then testing against $Y \in \mathfrak{z}(\mathfrak{g})$ yields $\langle Z, Y \rangle_{\mathfrak{z}} = 0$ for all $Y$, hence $Z = 0$; testing against $Y \in [\mathfrak{g}, \mathfrak{g}]$ yields $B(W, Y) = 0$ for all $Y$, hence $W = 0$ by the non–degeneracy of $B$ on the semisimple ideal. Thus $X = 0$. $\square$

**Proposition B.3** (Ad–invariance on the identity component). $\widetilde{B}$ is ad–invariant:

$$\widetilde{B}([X, Y], Z) + \widetilde{B}(Y, [X, Z]) = 0 \qquad \text{for all } X, Y, Z \in \mathfrak{g}, \tag{20}$$

and hence $\widetilde{B}(\mathrm{Ad}_g Y, \mathrm{Ad}_g Z) = \widetilde{B}(Y, Z)$ for all $g$ in the identity component $G^\circ$.

*Proof.* The restriction to $[\mathfrak{g}, \mathfrak{g}]$ equals $B$, which is ad–invariant. If $Z \in \mathfrak{z}(\mathfrak{g})$ then $[X, Z] = 0$ for all $X$, so any bilinear form on $\mathfrak{z}(\mathfrak{g})$ is automatically ad–invariant. Using Proposition B.1 and bilinearity gives the displayed identity. Equivalence with Ad–invariance on $G^\circ$ follows by integrating the infinitesimal relation along paths in $G^\circ$. $\square$

*Remark* 2 (Invariance for nonconnected groups). In case the group is nonconnected, and one desires invariance under the full group $G$ (not just $G^\circ$). The component group $\Gamma = G/G^\circ$ acts linearly on $\mathfrak{z}(\mathfrak{g})$. In all practical cases, $\Gamma$ will be a finite group. Then averaging any positive–definite $\langle \cdot, \cdot \rangle_{\mathfrak{z}}$ over $\Gamma$ yields an $\mathrm{Ad}(G)$–invariant inner product on the center:

$$\langle Z_1, Z_2 \rangle_{\mathfrak{z}}^{\mathrm{avg}} = \frac{1}{|\Gamma|} \sum_{\gamma \in \Gamma} \big\langle \mathrm{Ad}_\gamma Z_1, \ \mathrm{Ad}_\gamma Z_2 \big\rangle_{\mathfrak{z}}. \tag{21}$$

Replacing $\langle \cdot, \cdot \rangle_{\mathfrak{z}}$ by $\langle \cdot, \cdot \rangle_{\mathfrak{z}}^{\mathrm{avg}}$ in 18 makes $\widetilde{B}$ invariant under all of $G$.

## C  CONNECTIONS TO EXISTING BILINEAR FORMS

We demonstrate how the trace-based form $\widetilde{B}$ (Eq. 2) unifies and recovers prior bilinear constructions in the regimes used by Lie Neurons (Lin et al., 2024a) and Vector Neurons (Deng et al., 2021) . The discussion below states precise conditions under which $\widetilde{B}$ (i) equals the Killing-form contractions on semisimple inputs and (ii) is proportional to the Vector Neuron inner product under the $\mathfrak{so}(3) \simeq \mathbb{R}^3$ isomorphism. Our single form applies on the full reductive algebra $\mathfrak{gl}(n)$ including these specialized approaches, which encompasses general $n \times n$ matrix-valued inputs.

### C.1  REDUCTION TO THE KILLING FORM ON THE SEMISIMPLE IDEAL

Write $X = X_0 + \frac{1}{n}\mathrm{tr}(X)I$ with $X_0 \in \mathfrak{sl}(n)$. Using Eq. 2 we obtain

$$\widetilde{B}(X, Y) = 2n\,\mathrm{tr}(X_0 Y_0) + n^2\big(\tfrac{1}{n}\mathrm{tr}X\big)\big(\tfrac{1}{n}\mathrm{tr}Y\big). \tag{22}$$

Hence, when inputs are restricted to the semisimple ideal $[\mathfrak{g}, \mathfrak{g}] = \mathfrak{sl}(n)$ (so $\mathrm{tr}(X) = \mathrm{tr}(Y) = 0$), the center contribution vanishes and

$$\widetilde{B}|_{\mathfrak{sl}(n)}(X, Y) = 2n \cdot \mathrm{tr}(XY) = B_{\mathfrak{sl}(n)}(X, Y), \tag{23}$$

i.e. $\widetilde{B}$ coincides with the (scaled) Killing form used in Lie Neurons. More generally, for any semisimple subalgebra $\mathfrak{h} \subset [\mathfrak{g}, \mathfrak{g}]$, the restriction $\widetilde{B}|_{\mathfrak{h}}$ matches the Killing-form contraction on $\mathfrak{h}$ up to global scaling.

## C.2   $\mathfrak{so}(3)$ EXAMPLE: RECOVERY OF THE VECTOR NEURON INNER PRODUCT

Recall the hat map $\widehat{(\cdot)} : \mathbb{R}^3 \to \mathfrak{so}(3)$ with

$$\widehat{v} = \begin{bmatrix} 0 & -v_3 & v_2 \\ v_3 & 0 & -v_1 \\ -v_2 & v_1 & 0 \end{bmatrix}, \qquad \mathrm{tr}(\widehat{v}\,\widehat{w}) = -2\,v^\top w. \tag{24}$$

For $X = \widehat{v}$ and $Y = \widehat{w}$, Eq. 2 gives

$$\widetilde{B}(X, Y) = 2n \cdot \mathrm{tr}(XY) = 2n \cdot (-2\,v^\top w) = (-4n)\,v^\top w, \tag{25}$$

i.e. $\widetilde{B}$ is proportional to the Euclidean inner product on $\mathbb{R}^3$. The proportionality constant depends only on $n$; in practice this constant is absorbed by adjacent learnable linear layers or normalization, yielding behaviour identical to the inner product used in Vector Neurons to harmless scaling.

These remarks justify using the single, nondegenerate $\widetilde{B}$ across heterogeneous input types while preserving compatibility with prior architectures.

## D   DETAILED LAYER FORMULATIONS

This section provides the precise mathematical definitions and equivariance proofs for the core components of the ReLN architecture. We consider the input to a layer as a tensor $x \in \mathbb{R}^{K \times C}$, representing $C$ feature channels where $K = \dim \mathfrak{g}$. Each column $x_c \in \mathbb{R}^K$ is the vector representation of a feature. We use the wedge ($\wedge$) and vee ($\vee$) operators to map between the vector form $x_c$ and the matrix form $X_c \in \mathfrak{g}$.

### D.1   EQUIVARIANT LINEAR LAYER

The ReLN-Linear layer applies a linear map to the channel dimension of the input tensor $x$:

$$f_{\mathrm{ReLN-Lin}}(x; W) = xW, \quad \text{where } W \in \mathbb{R}^{C \times C'}. \tag{26}$$

We omit any bias term to preserve exact equivariance.

**Proof of Equivariance.** The group action, $\mathrm{Ad}_g$ (defined in Equation 11), is a linear map that multiplies each feature channel from the left. The weight matrix $W$ multiplies the channel dimension from the right. These operations commute, ensuring strict $G$-equivariance for any $g \in G$:

$$\begin{aligned} f_{\mathrm{ReLN-Lin}}(\mathrm{Ad}_g(x); W) &= (\mathrm{Ad}_g x)W \\ &= \mathrm{Ad}_g(xW) \\ &= \mathrm{Ad}_g(f_{\mathrm{ReLN-Lin}}(x; W)). \end{aligned} \tag{27}$$

### D.2   EQUIVARIANT NONLINEARITIES

Standard pointwise activations break equivariance under non-orthogonal transforms. We introduce two equivariant alternatives.

**ReLN-ReLU.** This layer rectifies a feature based on its alignment with a learnable direction. Given the input tensor $x$, we first compute per-channel vector directions $d = xU$. The nonlinearity for the input $x$ is then defined as:

$$f_{\mathrm{ReLN-ReLU}}(x) = \begin{cases} x, & \text{if } \widetilde{B}(x^\wedge, d^\wedge) \le 0, \\ x + \widetilde{B}(x^\wedge, d^\wedge)d, & \text{otherwise.} \end{cases} \tag{28}$$

Since all operations—the linear map to compute $d_i$, the bilinear form $\widetilde{B}$, and vector addition/scaling—are equivariant, the entire function is equivariant. The leaky variant $f_{\mathrm{ReLN-LeakyReLU}}(x) = \alpha\,x + (1 - \alpha)\,f_{\mathrm{ReLN-ReLU}}(x)$ follows directly.

**ReLN-Bracket (Lie-bracket nonlinearity).**  This layer uses the matrix commutator, a natural $\mathrm{Ad}$-equivariant primitive, to create learnable interactions between channels. Let the input be a batch of features represented by their vector coordinates, $x \in \mathbb{R}^{B \times C_{\text{in}} \times K}$. First, two independent linear maps (with learnable weights $W_a, W_b \in \mathbb{R}^{C_{\text{in}} \times C_{\text{out}}}$) transform the input features into two intermediate tensors, $u, v \in \mathbb{R}^{B \times C_{\text{out}} \times K}$. The Lie bracket is then computed channel-wise between the corresponding feature vectors of $u$ and $v$. This produces an update tensor, $\Delta x \in \mathbb{R}^{B \times C_{\text{out}} \times K}$, where each vector is defined as:

$$(\Delta x)_{b,c',:} = [(u_{b,c',:})^\wedge, (v_{b,c',:})^\wedge]^\vee. \tag{29}$$

This update is added to the input for a residual connection (requiring $C_{\text{in}} = C_{\text{out}}$ for the shapes to match):

$$f_{\text{ReLN}-\text{Bracket}}(x) = x + \Delta x. \tag{30}$$

Each step in this process (linear map, the Lie bracket, and vee/hat operations) is equivariant under the adjoint action, making the entire block equivariant $f_{\text{ReLN}-\text{Bracket}}(\mathrm{Ad}_g x) = \mathrm{Ad}_g(f_{\text{ReLN}-\text{Bracket}}(x))$ for all $g \in G$.

# E  Experimental Details

Training and evaluation for all presented experiments, Platonic solid classification, invariant function regression, top tagging, and drone state estimation, were conducted on a single NVIDIA GeForce RTX 4090 GPU.

## E.1  Model Architectures and Implementation Details

Across all experiments, our proposed ReLN models are constructed by stacking ReLN-Linear, ReLN-ReLU, and ReLN-Bracket layers. The specific number of layers and channel widths are adapted for each task to ensure a fair comparison with baseline models in terms of parameter count.

**Algebraic Benchmarks ($\mathfrak{sl}(3)$ and $\mathfrak{sp}(4)$).**  For the Platonic solid classification and $\mathfrak{sp}(4)$ invariant regression tasks, our ReLN model directly adopts the architecture used by the Lie Neurons benchmark model from Lin et al. (2024a). The primary modification is the replacement of their Killing form-based nonlinearity and invariant layers with our proposed non-degenerate bilinear form $\widetilde{B}$ (Eq. 2). This setup allows for a direct comparison of the impact of the bilinear form, as all other architectural hyperparameters are kept identical to the baseline.

**Top Tagging.**  For the Top-Tagging task, our model is a modification of the LorentzNet architecture (Gong et al., 2022). We adapt its Lorentz Group Equivariant Blocks (LGEBs) by replacing the invariant feature computation with our proposed bilinear form. A detailed description of the architecture, our modifications, and training protocol is provided in Appendix F.

**Drone State Estimation.**  In this task, we compare our ReLN model against two baseline families: a non-equivariant 1D ResNet and an equivariant Vector Neurons (VN) model. The specific implementation details and architectural choices for each model are provided next in Appendix G.

## E.2  Platonic Solid Classification on $\mathfrak{sl}(3)$

**Overview.**  All experiments evaluate classification of Platonic solids (tetrahedron, octahedron, icosahedron) from inter-face homographies computed in the image plane. For each model-family we train 5 independent runs with different random seeds and report mean ± standard deviation. Training uses fixed object and camera poses; at test time we report results on the in-distribution (ID) split and the rotated-camera (RC) split (RC applies ten random $\mathrm{SO}(3)$ rotations to the camera frame). The 'MLP (wider)' denotes a capacity-matched ($\approx 2\times$ parameters) MLP used for a fairer comparison.

Table 6: Common training hyperparameters (used across model families unless noted).

| Hyperparameter | Value |
|---|---|
| Optimizer | Adam |
| Batch size | 100 |
| Number of independent runs (seeds) | 5 |
| Max epochs / stopping criterion | 5000 epochs |
| Data augmentation (train) | Random camera rotations applied to training examples when enabled |
| RC evaluation | 500 random $SO(3)$ rotations applied to each test example |
| Metric reported | Classification accuracy (mean ± std across runs) |

Table 7: Model-specific hyperparameters and implementation notes.

| Model family | Key choices | Notes |
|---|---|---|
| Latent Feature Size (MLP Baseline) | 256 | As in Lin et al. (2024a). |
| Latent Feature Size (MLP Wider) | 386 | Increased width total parameters $\approx 2\times$ baseline. |
| Learning rate (MLP models) | $1 \times 10^{-4}$ | |
| Learning rate (ReLNs/Lie Neurons models) | $3 \times 10^{-6}$ | Lower LR chosen for stable training |

# F  TOP TAGGING EXPERIMENT: FRAMEWORK, PROOF, AND IMPLEMENTATION

This appendix provides the complete details for our jet tagging experiment. We first present the geometric framework and the mathematical proof of our Lorentz-equivariant embedding, and then describe the model architecture and training protocol.

## F.1  GROUP ACTION EQUIVARIANCE VIA EMBEDDING MAP

To process four-momenta within our Lie-algebraic framework, we require an embedding that translates the action of the Lorentz group into an adjoint action on a matrix space. This is achieved by lifting the four-vector into $\mathfrak{gl}(5)$.

**Definition F.1** (Lorentz-Compatible Embedding). Given a four-vector $p \in \mathbb{R}^4$ and the Minkowski metric $\eta = \mathrm{diag}(1, -1, -1, -1)$, we define its embedding $\varphi(p)$ into $\mathfrak{gl}(5)$ as:

$$\varphi(p) = \begin{bmatrix} 0_{4\times 4} & p \\ p^\top \eta & 0 \end{bmatrix}. \tag{31}$$

**Theorem F.1** (Adjoint Equivariance). *The embedding $\varphi$ correctly models the Lorentz group action. For any $p \in \mathbb{R}^4$ and Lorentz transformation $\Lambda \in SO(1,3)$, let $G = \mathrm{diag}(\Lambda, 1) \in GL(5)$. The map is equivariant in the sense that the standard action on $p$ corresponds to the adjoint action on $\varphi(p)$:*

$$\mathrm{Ad}_G(\varphi(p)) = G\varphi(p)G^{-1} = \varphi(\Lambda p). \tag{32}$$

*Proof.* We compute the left-hand side (LHS) of Eq. 32, which is the adjoint action:

$$\mathrm{Ad}_G(\varphi(p)) = \begin{bmatrix} \Lambda & 0 \\ 0 & 1 \end{bmatrix} \begin{bmatrix} 0 & p \\ p^\top \eta & 0 \end{bmatrix} \begin{bmatrix} \Lambda^{-1} & 0 \\ 0 & 1 \end{bmatrix} = \begin{bmatrix} 0 & \Lambda p \\ p^\top \eta \Lambda^{-1} & 0 \end{bmatrix}. \tag{33}$$

The right-hand side (RHS) is the lift of the transformed vector $\Lambda p$:

$$\varphi(\Lambda p) = \begin{bmatrix} 0 & \Lambda p \\ (\Lambda p)^\top \eta & 0 \end{bmatrix} = \begin{bmatrix} 0 & \Lambda p \\ p^\top \Lambda^\top \eta & 0 \end{bmatrix}. \tag{34}$$

For the LHS and RHS to be equal, we must show that $\eta \Lambda^{-1} = \Lambda^\top \eta$. We start from the defining property of $SO(1,3)$:

$$\Lambda^\top \eta \Lambda = \eta. \tag{35}$$

Right-multiplying Eq. 35 by $\Lambda^{-1}$ yields the desired identity:

$$(\Lambda^\top \eta \Lambda)\Lambda^{-1} = \eta \Lambda^{-1} \implies \Lambda^\top \eta (\Lambda \Lambda^{-1}) = \eta \Lambda^{-1} \implies \Lambda^\top \eta = \eta \Lambda^{-1}. \tag{36}$$

Since the condition holds, the proof is complete. $\square$

*Remark* 3 (Generalization to Orthogonal Groups). This embedding technique is not limited to the Lorentz group and can be readily generalized to any orthogonal group $O(n)$ or special orthogonal group $SO(n)$. For instance, in applications involving 3D point clouds where the symmetry is $SO(3)$, a vector $p \in \mathbb{R}^3$ would be embedded into the Lie algebra $\mathfrak{gl}(4)$ as:

$$\varphi(p) = \begin{bmatrix} 0_{3\times 3} & p \\ p^\top & 0 \end{bmatrix} \tag{37}$$

The proof of equivariance follows the same structure, using the property of orthogonal matrices, $R^\top R = I$ (which implies $R^{-1} = R^\top$), instead of the Minkowski metric identity. This highlights the broad applicability of our embedding strategy to any benchmark involving norm-preserving group transformations.

### F.2 EXPERIMENTAL IMPLEMENTATION

**Dataset**  The experiment uses the Top-Tagging dataset (Kasieczka et al., 2019), which contains 2 million simulated proton-proton collision events. The dataset was generated with Pythia, Delphos, and FastJet to model the ATLAS detector response. We use the standard 60%/20%/20% splits for training, validation, and testing. Each jet is represented as a set of constituent particles, each with four-momentum $p = (E, p_x, p_y, p_z)$.

**Model**  Our model leverages the established architecture of LorentzNet (Gong et al., 2022), utilizing its stack of Lorentz Group Equivariant Blocks (LGEBs) for message passing on the jet's particle cloud. While the original LorentzNet computes these features directly from the 4-momenta using the Minkowski inner product, our approach introduces a modified bilinear form based feature extraction. We first embed each pair of 4-momenta, $p_i$ and $p_j$, from the Minkowski space $\mathbb{R}^{1,3}$ into the Lie algebra $\mathfrak{gl}(5)$ via the map $p \mapsto X(p)$. The invariant features for the message passing are then derived from the bilinear form, $B(\cdot, \cdot)$, on this Lie algebraic space. The edge message $m_{ij}$ is thus constructed as:

$$m_{ij} = \phi_e\Big(h_i, h_j, \psi\big(\widetilde{B}(X(p_i), X(p_i))\big), \psi\big(\widetilde{B}(X(p_i), X(p_j))\big)\Big) \tag{38}$$

where $h_i, h_j$ are scalar features, $\phi_e$ is an MLP, and $\psi$ is a stabilizing nonlinearity. As shown in the main results (Table 4), this approach leads to an advantage in background rejection when compared against a parameter-matched LorentzNet baseline. The architectural differences are summarized in Table 8.

Table 8: Architectural comparison for the Top-Tagging task.

| Component | LorentzNet (Original) | Param-matched Baseline | Ours (ReLN) |
|---|---|---|---|
| Number of LGEBs | 6 | 5 | 5 |
| Hidden feature dims | 72 | 48 | 48 |
| Edge feature computation | Minkowski inner prod. | Minkowski inner prod. | **Bilinear invariant form** |

**Training Setup**  For a fair comparison, our training procedure closely follows the protocol established in the LorentzNet (Gong et al., 2022). The model was trained for a total of 35 epochs on a single NVIDIA RTX 4090 GPU. We used the AdamW optimizer with a weight decay of 0.01 and a batch size of 128, matching the total effective batch size from the reference work. The learning rate was managed by the paper's specific three-stage schedule: a 4-epoch linear warm-up to an initial rate of $1 \times 10^{-3}$, followed by a 28-epoch `CosineAnnealingWarmRestarts` schedule, and a final 3-epoch exponential decay. After each epoch, the model with the highest validation accuracy was saved for final evaluation on the test set.

## G  DRONE EXPERIMENT DETAILS

This appendix provides the technical details for the drone state estimation experiment, including the theoretical framework, dataset generation, model implementations, and formal proofs.

### G.1 GEOMETRIC FRAMEWORK FOR EQUIVARIANT COVARIANCE PROCESSING

Our approach leverages the geometry of symmetric positive-definite matrices. A covariance matrix $C$ is symmetric positive-definite, residing on the manifold $\mathrm{SPD}(3)$. A non-degenerate covariance matrix $C \in \mathrm{SPD}(n)$ represents the anisotropic stretching of a general linear map, as seen via the polar decomposition $A = QP$ with $Q \in \mathrm{O}(n)$ and $P \in \mathrm{SPD}(n)$. Equivalently, there is a homogeneous-space isomorphism: $\mathrm{SPD}(n) \cong \mathrm{GL}(n)/\mathrm{O}(n)$, which motivates processing covariances in a $\mathrm{GL}(n)$-aware architecture.

While $\mathrm{SPD}(3)$ is not a Lie group, the matrix logarithm provides a canonical map to the vector space of symmetric matrices $\mathrm{Sym}(3)$, which is a linear subspace of $\mathfrak{gl}(3)$.

$$\log : \mathrm{SPD}(3) \longrightarrow \mathrm{Sym}(3) \subset \mathfrak{gl}(3). \tag{39}$$

This allows us to embed a geometric object from a curved manifold into a flat, Lie-algebra-compatible space. The following theorem proves that the congruence transformation on $C \in \mathrm{SPD}(n)$ becomes an adjoint action on its image $\log C \in \mathrm{Sym}(n)$, thus preserving the equivariant structure required by our model.

**Theorem G.1** (Equivariance of the Logarithmic Map). *For any $C \in \mathrm{SPD}(n)$ and any rotation matrix $R \in \mathrm{SO}(n)$, the congruence transformation on $C$ corresponds to an adjoint action on its logarithm:*

$$\log(RCR^\top) = R(\log C)R^\top. \tag{40}$$

*Proof.* The proof follows from the spectral theorem for real symmetric matrices.

1. Let the eigendecomposition of $C$ be $C = V\Lambda V^\top$, where $V$ is an orthogonal matrix ($V^\top V = I$) of eigenvectors and $\Lambda$ is the diagonal matrix of corresponding positive eigenvalues.

2. By definition, the matrix logarithm of $C$ is given by applying the logarithm to its eigenvalues:
$$\log C := V(\log \Lambda)V^\top \tag{41}$$
where $\log \Lambda$ is the diagonal matrix of element-wise logarithms of the eigenvalues.

3. Consider the transformed matrix $C' = RCR^\top$. Substituting the decomposition of $C$ yields:
$$C' = R(V\Lambda V^\top)R^\top = (RV)\Lambda(V^\top R^\top) = (RV)\Lambda(RV)^\top \tag{42}$$
This is the eigendecomposition of $C'$, where the new orthogonal matrix of eigenvectors is $V' = RV$ and the eigenvalues $\Lambda$ are unchanged.

4. Applying the definition of the matrix logarithm to $C'$ gives:
$$\log(C') = V'(\log \Lambda)(V')^\top = (RV)(\log \Lambda)(RV)^\top \tag{43}$$

5. Rearranging the terms, we arrive at the desired identity:
$$\log(C') = R\left(V(\log \Lambda)V^\top\right)R^\top = R(\log C)R^\top \tag{44}$$

$\square$

This identity is critical, as it confirms that our adjoint-equivariant network can process either the raw covariance $C$ or its logarithm $\log C$ while perfectly preserving the $\mathrm{SO}(3)$ symmetry.

In the $\mathrm{SO}(3)$ regime used in our experiments, vectors (e.g., velocity $\mathbf{v}$) are represented in the Lie algebra $\mathfrak{so}(3)$ so that the adjoint action coincides with ordinary rotation, $\mathrm{Ad}_R(\mathbf{v}) = R\mathbf{v}$. Conjugation then implements the covariance congruence $C \mapsto RCR^\top$. Consequently, ReLNs realize $\mathrm{SO}(3)$-equivariance *by construction*, avoiding the need for the model to learn these symmetries from data.

### G.2 DATASET GENERATION.

We use the PyBullet engine to simulate 200 aggressive trajectories for a Crazyflie-like nano-quadrotor. To generate realistic measurements, the instantaneous velocity is corrupted by Gaussian noise, $\mathbf{v}_{\mathrm{noisy}} \sim \mathcal{N}(\mathbf{v}_{\mathrm{gt}}, C_v)$, where the covariance $C_v$ varies with flight aggressiveness. The dataset provides time series of noisy velocities, ground-truth covariances, and ground-truth trajectories for evaluation.

**Trajectory Generation.** The procedure begins with the procedural generation of a sequence of 20 to 40 random 3D waypoints within a flight volume of approximately $170\text{m} \times 170\text{m} \times 60\text{m}$. The waypoints are sampled from a uniform distribution to create diverse flight paths. To mimic the complex dynamics of aggressive flight, each trajectory is randomly generated using a path with random wiggles or a path featuring high-speed spiral maneuvers. These discrete waypoints are then interpolated using a Catmull-Rom spline to create a smooth, $C^1$ continuous target trajectory, which is densely sampled at an 80 Hz control frequency. Each of the 200 sequences results in a unique trajectory lasting approximately 2-4 minutes, totaling over 13 hours of simulated flight time. A sample generated trajectory is shown in Figure 5.

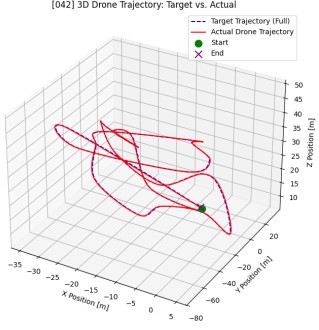 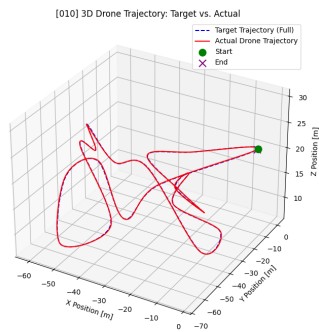

(a) A sample trajectory with spiral maneuvers.    (b) A sample trajectory with random wiggles.

Figure 5: Sample aggressive trajectories generated in the PyBullet simulator.

**State-Dependent Noise Model.** To simulate realistic sensor characteristics, the ground-truth velocity is corrupted by zero-mean Gaussian noise, $\mathbf{v}_{\text{noisy}} \sim \mathcal{N}(\mathbf{v}_{\text{gt}}, C_v)$. The covariance matrix $C_v$ is state-dependent, designed to scale with the drone's speed. The standard deviation $\sigma_v$ for each velocity axis is computed using a sigmoid function of the velocity magnitude $\|\mathbf{v}_{\text{gt}}\|$:

$$\sigma_v(\|\mathbf{v}_{\text{gt}}\|) = \sigma_{\min} + (\sigma_{\max} - \sigma_{\min}) \cdot \frac{1}{1 + \exp(-\lambda(\|\mathbf{v}_{\text{gt}}\| - v_{\text{mid}}))}, \tag{45}$$

where the variance on each axis is $\sigma_v^2$. We set the minimum and maximum standard deviations to $\sigma_{\min} = 0.2\,\text{m/s}$ and $\sigma_{\max} = 1.0\,\text{m/s}$, respectively. The steepness $\lambda$ is set to 0.8, and the midpoint velocity $v_{\text{mid}}$ is dynamically adjusted based on the estimated average speed of each trajectory to ensure a realistic noise profile.

### G.3 BASELINE AND MODEL IMPLEMENTATION DETAILS

We compare ReLN against two baseline classes chosen to isolate the effect of geometric priors.

**Non-equivariant baselines.** We use a standard 1D ResNet architecture with temporal convolutional blocks that processes flattened input sequences. The **ResNet (velocity-only)** model receives only the 3D velocity vector. The **ResNet (velocity + covariance)** model receives the flattened $3 \times 3$ covariance matrix concatenated to the velocity vector.

**Eigendecomposition-based** $SO(3)$**-Equivariant Baseline.** This model adapts the 1D ResNet backbone for $SO(3)$ equivariance using VN layers. Since VNs cannot directly ingest matrices, we decompose each covariance matrix $C = V\Lambda V^\top$ and use a dual-stream design:

- an *equivariant* stream $\mathcal{F}_{\text{eq}} = \{\mathbf{v}, \mathbf{e}_1, \mathbf{e}_2, \mathbf{e}_3\}$ comprising the measured velocity $\mathbf{v}$ and the three orthonormal eigenvectors $\mathbf{e}_i$, which together capture all directional information. This stream is handled by the VNs backbone.
- an *invariant* stream $\mathcal{F}_{\text{inv}} = \{\lambda_1, \lambda_2, \lambda_3\}$ processes the corresponding eigenvalues $\{\lambda_1, \lambda_2, \lambda_3\}$, which encode orientation-independent scale information, using a standard MLP.

The two latent features from both streams are fused at the final output layer. Eigenvector ambiguities (sign or multiplicities) are resolved via a deterministic, rotation-equivariant canonicalization.

**Reductive Lie Neurons (ReLNs).** The ReLN model shares the same base architecture as the VN model but incorporates ReLN-Bracket layer as an additional source of nonlinearity after the initial feature extraction block. In contrast to the VN backbone, ReLNs provide a unified framework for velocity and covariance processing. Velocities $\mathbf{v} \in \mathbb{R}^3$ are lifted into the Lie algebra $\mathfrak{so}(3)$ via $K = \mathbf{v}^\wedge$, while covariance $C$ (or $\log C$) is treated as a structured geometric input. Both transform under the same adjoint action: $K' = RKR^\top$ and $C' = RCR^\top$, enabling joint equivariant processing. Although $\mathrm{SPD}(n)$ is *not* a Lie algebra or group, it is a subset of $\mathrm{GL}(n)$. By taking the matrix logarithm, $\log C \in \mathrm{Sym}(n) \subset \mathfrak{gl}(n)$, covariances are embedded into a linear subspace compatible with Lie-algebra processing.

The network $\Phi$ fuses these inputs into a single matrix $A \in \mathbb{R}^{3\times 3}$, from which we extract the velocity estimate:

$$A_{\mathrm{skew}} := \tfrac{1}{2}(A - A^\top) \in \mathfrak{so}(3), \qquad \hat{\mathbf{v}} = \mathrm{Vee}(A_{\mathrm{skew}}). \qquad (46)$$

The extracted velocity is provably $\mathrm{SO}(3)$-equivariant by construction. See Appendix H for the full statement and proof.

## G.4 Training and Evaluation Protocol

**Problem Formulation** The network is trained to predict the drone's 3D position $\mathbf{p}_t \in \mathbb{R}^3$ at the end of a given time window, based on a sequence of noisy velocity measurements and their corresponding covariances within that window (e.g., a 1-second history). All models are trained by minimizing the Mean Squared Error (MSE) between the predicted position $\hat{\mathbf{p}}_t$ and the ground-truth position $\mathbf{p}_{t,\mathrm{gt}}$. The loss function is defined as $\mathcal{L} = \|\hat{\mathbf{p}}_t - \mathbf{p}_{t,\mathrm{gt}}\|_2^2$.

**Dataset and Optimization.** We partition the dataset using a standard 80:10:10 train/validation/test split. All models are trained on identical splits to ensure fair comparison. Models are optimized using the AdamW optimizer with a ReduceLROnPlateau learning rate scheduler based on validation loss.

**Evaluation Metrics.** We report the following pose-regression metrics over the test set:

- **Absolute Trajectory Error (ATE):** The root-mean-square error between the ground-truth and predicted 3D positions over the entire trajectory, measured in meters.
- **ATE$_\%$:** The ATE normalized by the total trajectory length and expressed as a percentage ($100 \times \mathrm{ATE}/\mathrm{length}$). This metric provides a scale-invariant measure of error, which is crucial for fairly comparing performance across our aggressive flight trajectories of varying lengths.
- **Relative Pose Error (RPE):** The error measured over fixed-length sub-trajectories, capturing local drift.

To explicitly validate equivariance, we also evaluate all models on the test set after applying a set of random $\mathrm{SO}(3)$ rotations to the entire input sequence.

## G.5 Eigenvector Canonicalization for the VN Baseline

To resolve ambiguities in the eigendecomposition $C = V\Lambda V^\top$ for the VN baseline, we canonicalize the eigenvector matrix $V = [\mathbf{e}_1, \mathbf{e}_2, \mathbf{e}_3]$ as follows:

1. **Right-handed Frame:** If $\det V < 0$, we set $\mathbf{e}_3 \leftarrow -\mathbf{e}_3$ to ensure $\det V = +1$.
2. **Sign Disambiguation:** For each eigenvector $\mathbf{e}_i$ with a distinct eigenvalue, we enforce a consistent sign by ensuring $\mathbf{v}^\top \mathbf{e}_i \geq 0$. If not, we set $\mathbf{e}_i \leftarrow -\mathbf{e}_i$.
3. **Multiplicity Handling:** In the rare case of repeated eigenvalues, we use the projection of the velocity vector $\mathbf{v}$ onto the corresponding eigenspace to deterministically define the first basis vector, then complete the basis via Gram-Schmidt.

All steps use only equivariant operations, preserving the overall symmetry of the baseline.

# H    PROOF OF SO(3)-EQUIVARIANCE FOR ReLN VELOCITY EXTRACT WITH COVARIANCE INPUTS

This section provides a formal proof for the SO(3)-equivariance of our Reductive Lie Neuron (ReLN) architecture when processing a velocity vector and a covariance matrix. We first establish the foundations for processing covariance matrices within a Lie-algebraic framework and then present the main proof.

## H.1    SO(3)-EQUIVARIANT VECTOR EXTRACTION VIA SKEW-SYMMETRIC PROJECTION

Our network, $\Phi$, is designed to be adjoint-equivariant. It maps geometric inputs—such as a embedded velocity $K \in \mathfrak{so}(3)$ and a covariance matrix $S \in \mathrm{SPD}(3)$—to a matrix feature $A \in \mathbb{R}^{3\times3}$. The inputs transform under the adjoint action of any rotation $R \in \mathrm{SO}(3)$:

$$K' = \mathrm{Ad}_R(K) = RKR^\top, \quad S' = \mathrm{Ad}_R(S) = RSR^\top. \tag{47}$$

By construction, the network's output feature $A$ transforms according to the same law:

$$\Phi(K', S') = \mathrm{Ad}_R\big(\Phi(K, S)\big) = R\,\Phi(K, S)\,R^\top. \tag{48}$$

To obtain the final 3D velocity vector, we project the output matrix $A$ onto its skew-symmetric component and apply the vee operator. The following proposition formalizes the equivariance of this extraction mechanism.

**Proposition H.1** (Equivariance of Skew-Symmetric Extraction)**.** Let a network $\Phi$ and its inputs transform according to Eqs. 47 and 48. If a vector $\hat{\mathbf{v}} \in \mathbb{R}^3$ is extracted from the output matrix $A = \Phi(K, S)$ via the projection

$$A_{\mathrm{skew}} = \tfrac{1}{2}(A - A^\top), \qquad \hat{\mathbf{v}} = (A_{\mathrm{skew}})^\vee, \tag{49}$$

then the vector $\hat{\mathbf{v}}'$ extracted from the transformed output $A' = \Phi(K', S')$ transforms covariantly as $\hat{\mathbf{v}}' = R\hat{\mathbf{v}}$.

*Proof.* By the adjoint-equivariance property in Eq. 48, the network satisfies $\Phi(RKR^\top, RCR^\top) = R\,\Phi(K, C)\,R^\top = RAR^\top$. Let $A' = RAR^\top$. The skew-symmetric component of the transformed output $A'$ is:

$$\begin{aligned}
A'_{\mathrm{skew}} &= \tfrac{1}{2}(A' - A'^\top) \\
&= \tfrac{1}{2}\big(RAR^\top - (RAR^\top)^\top\big) \\
&= \tfrac{1}{2}\big(RAR^\top - RA^\top R^\top\big) \\
&= R\left(\tfrac{1}{2}(A - A^\top)\right)R^\top \\
&= RA_{\mathrm{skew}}R^\top = \mathrm{Ad}_R(A_{\mathrm{skew}}).
\end{aligned} \tag{50}$$

The vee map, $(\cdot)^\vee : \mathfrak{so}(3) \to \mathbb{R}^3$, is itself an equivariant map satisfying $(\mathrm{Ad}_R(X))^\vee = R\,(X^\vee)$ for any $X \in \mathfrak{so}(3)$. Applying this property yields the desired result:

$$\hat{\mathbf{v}}' = (A'_{\mathrm{skew}})^\vee = (\mathrm{Ad}_R(A_{\mathrm{skew}}))^\vee = R\,(A_{\mathrm{skew}})^\vee = R\hat{\mathbf{v}}. \tag{51}$$

$\square$

*Remark* 4. The proof relies on three properties: (i) both inputs transform under the adjoint action $X \mapsto RXR^\top$; (ii) the network $\Phi$ is equivariant to this action; and (iii) the output is projected onto $\mathfrak{so}(3)$ before the vee operator is applied. As established previously, these conditions hold whether the network ingests the raw covariance $S$ or its logarithm $\log S$.

## LARGE LANGUAGE MODEL (LLM) USAGE

We used a large language model (LLM) to aid in polishing the writing and improving grammatical clarity of the manuscript. The LLM did not contribute to the research ideation, experiments, or technical content; all scientific claims and results were generated solely by the authors.

