# OpenReview forum: "Equivariant Neural Networks for General Linear Symmetries on Lie Algebras"
_ICLR.cc/2026/Conference — ICLR 2026 Conference Desk Rejected Submission_

### Official Review · Reviewer_psZr · 2025-10-31

**Soundness:** 4
**Presentation:** 2
**Contribution:** 3
**Rating:** 4
**Confidence:** 4

**Summary:**

The paper proposes a novel equivariant architecture framework "Reductive Lie Neuron" (ReLN) which generalizes the "Lie Neuron" architecture from Lin et al. 2024 from semisimple Lie groups to reductive Lie groups.
This is achieved by considering a more general and expressive bilinear form instead of the Killing form, which can be degenerate on reductive Lie algebras.

**Strengths:**

The proposed idea is elegant and interesting, and can be potentially impactful in some of the applications described by the authors.
The manuscript is quite well written, although it provides little background on the advanced mathematical tools leveraged, which might make it less accessible to the general audience (see comments below).

**Weaknesses:**

While the paper is well motivated and mentions some exciting applications, I found the experimental validation less interesting and weaker.
Indeed, the authors consider 4 tasks, 2 simple synthetic datasets with *semisimple* Lie Algebras in Sec 5.1, an apparently saturated Jet-Tagging benchmark with the reductive gl(n) Lie group in Sec 5.2, and (if I understood correctly) a SO(3) equivariant task in section 5.3.

While these experiments seems good proof of concepts, I don't think they provide good arguments for the adoption of ReLN: in the first 2 tasks (Sec 5.1) ReLN simply matches Lie Neuron since the Lie algebra is semisimple, in the last one (Sec 5.3) the task is only SO(3) symmetric and the authors only consider an arguably poor SO(3) equivariant baseline. The Jet-Tagging task in Sec 5.2 seems to be the only one where reductive Lie Algebra equivariance is really necessary but the saturated benchmark doesn't show any strong benefits of ReLN with respect to less equivariant baselines.

In the conclusions, the authors claim the proposed method achieves SOTA results on benchmarks and deliver large gains in challenging drone state estimation task, but I am a bit more skeptical about these claims.
The authors suggest a wider array of physical systems, including dynamics of articulated robots and large scale sensor fusion as future applications to explore. These seems very interesting and exciting applications indeed, so I wonder why not benchmarking the proposed method on these relevant problems rather than the toy experiments considered in the manuscript.

See also questions below.

**Questions:**

I think it might be wroth spending a couple of words to explain what it is a reductive Lie algebra, the Killing form and what it means for it to be degenerate or, at least, the authors could point to some definition provided later.
For example, in Sec 4.1, different constructions based on these terms are described but no definition of reductive Lie algebra or Killing form is provided yet.
I am very familiar with the equivariance literature but not so much with reductive Lie groups, so I am not sure how to interpret exactly the equations in Definition 4.1.

Since the method builds on top of Lie Neurons (Line et al 2024a), it might be worth giving a quick overview of it first.


Eq. 4: is $d_c$ somehow dependent on the input too? Otherwise, if that is a fixed learnable vector, that operation doesn't seem equivariant, right?
Similar question for the Max-Killing pooling: is $D_{n,c}$ depending on the input?


Line 371: parameter matching is not really the most fair comparison. Why not doing a simple hyperparameter search and pick the best configuration for each model? E.g., from Table 4, it seems that the original LorentzNet had has 200K params, so why matching LorentzNet to the 84K params of ReLN rather than scaling ReLN to the 200K params of LorentzNet?

I am not very familiar with the Jet-Tagging dataset, but the results in Table 4 suggest this benchmark is quite saturated as most models achieve the exact same accuracy and AUC. Even the two instances of LorentzNet have the exact same performance despite the different number of params. Can the authors comment on this? Is there a better dataset the authors could validate the model on?


Sec 5.3: I might have misunderstood something, but isn't this task just E(3) equivariant? Indeed, in Table 5, only SO(3) test-time augmentation is performed. Then, why should we care about GL(n) here? I understand that the covariance matrices live in the SPD(3) manifold, but the group acting on them here is just SO(3) (or O(3)), so one could build a very competitive baseline using most of the related literature on E(3)-equivariant methods. If this is the case, then, why do you only consider a tweaked Vector-Neuron architecture as a baseline (which moreover seems to be one of the few related works which can't handle adjoint equivariance). Am I missing something?

---

> ### Author Response · Authors · 2025-11-21
>
> We thank the reviewer for their detailed and constructive feedback on both the theoretical and empirical aspects of our work. In this response, we clarify the role of $GL(n)$-equivariance, present additional results on articulated dynamics and uncertainty-aware estimation, and address the specific points raised.
>
> ---
>
> ### 1. Drone state estimation: reductive $\mathfrak{gl}(3)$ vs. SE(3)$ and Lie Neurons
>
> The drone task in Sec. 5.3 is exactly where the reductive $\mathfrak{gl}(3)$ structure matters. While the drone pose follows $SE(3)$, the model also processes (log-)covariance matrices describing uncertainty. These live on an SPD manifold and transform under congruence
>
> $\Sigma \mapsto A \Sigma A^\top,$
>
> which corresponds to the adjoint action of $GL(3)$ on $\mathfrak{gl}(3)$ and couples rotation with scaling/shear. This is precisely a setting where the Killing form is degenerate and our reductive construction is needed.
> To test this, we:
> - Added a strong $SE(3)$-equivariant baseline, SE(3)-Transformer [4];
> - Added “no-center” ReLN variants that are mathematically equivalent to Lie Neurons on $\mathfrak{sl}(3)$.
>
> **Table 1: Drone state estimation with (log-)covariance in $\mathfrak{gl}(3)$.**
>
> | Model               | Input type                     | ATE ↓  | ATE% ↓ | RPE ↓  |
> |---------------------|--------------------------------|-------:|-------:|-------:|
> | VN                  | Vel only                  | 17.36  | 7.52   | 13.51  |
> | VN                  | Vel + Cov          | 191.78 | 88.66  | 98.39  |
> | SE(3)-T   | Vel + Cov          | 16.45  | 7.09   | 12.85  |
> | SE(3)-T   | Vel + log-Cov      | 16.83  | 7.56   | 13.34  |
> | ReLN                | Vel only                  | 16.85  | 7.31   | 12.70  |
> | ReLN (full)         | Vel + Cov          | 16.49  | 7.21   | 13.02  |
> | **ReLN (full)**     | **Vel + log-Cov**  | **13.92** | **5.99** | **11.04** |
> | ReLN (no-center)\*  | Vel + Cov          | 16.86  | 7.43   | 13.65  |
> | ReLN (no-center)\*  | Vel + log-Cov      | 15.65  | 6.76   | 12.04  |
>
> \* “No-center” = Killing form restricted to $\mathfrak{sl}(3)$, trace direction ignored; **equivalent to the Lie Neurons setting**.
>
> **Key takeaways:**
> - **ReLN vs.SE(3)$-equivariant SE(3)-Transformer.**
>   Adding (log-)covariances to SE(3)-Transformer yields only minor changes (ATE $16.45 \to 16.83$), suggesting that treating covariances as generic feature tensors does not exploit SPD geometry. In contrast, **ReLN (full)** with log-covariances in $\mathfrak{gl}(3)$ improves ATE to **13.92** (≈15% better than SE(3)-Transformer), indicating that the reductive $\mathfrak{gl}(3)$ formulation is beneficial in practice.
> - **ReLN vs. semisimple Lie Neurons (no-center).**
>   The **no-center** variants, which coincide with Lie Neurons on $\mathfrak{sl}(3)$, consistently underperform full ReLN (e.g., ATE 15.65 vs.\ 13.92 with log-covariance). This shows that **ignoring the degenerate center harms performance** on uncertainty-aware dynamics and that the reductive extension is empirically important, not only a formal generalization.
> Thus, Sec. 5.3 demonstrates that full reductive $\mathfrak{gl}(3)$ modeling is what yields the observed gains.
>
> ---
>
> ### 2. Articulated dynamics: ReLN vs. EMLP on double pendulum
>
> To evaluate ReLN on a more complex articulated system, we added the **double-pendulum Hamiltonian dynamics benchmark** introduced by EMLP [1]. This is a standard test for matrix-group equivariant MLPs.
>
> **Accuracy.** ReLN **matches or slightly improves** test error compared to EMLP across all symmetry groups (O(2), SO(2), $D_6$) while maintaining exact equivariance:
>
> | Task | EMLP O(2) | **ReLN O(2)** | EMLP SO(2) | **ReLN SO(2)** | EMLP $D_6$ | **ReLN $D_6$** | MLP (Id) |
> | :--- | --------: | ------------: | ----------:| -------------: | -----------:| --------------:| -------: |
> | HNNs | 0.012(2)  | **0.011(2)**  | 0.015(3)   | **0.010(4)**   | 0.013(2)    | **0.011(2)**   | 0.028    |
>
> **Efficiency.** ReLN avoids per-layer numerical constraint solving for equivariant bases and is substantially cheaper in FLOPs:
>
> | Model        | #Params | FLOPs / step |
> | :----------- | ------: | -----------: |
> | MLP-HNN      | 34,817  | 70,400       |
> | EMLP-HNN [1] | 55,569  | 1,589,909    |
> | ReLN-HNN | 69,889  | 142,190 |
>
> On this canonical EMLP benchmark, ReLN therefore provides an **efficient alternative**: comparable or better accuracy at roughly **$11\times$** lower FLOPs per step.
>
> **Single $GL(n)$-equivariant backbone.**
> Since ReLN is $GL(n)$-equivariant by design, equivariance to any matrix subgroup $G \subset GL(n)$ acting by conjugation follows by restriction of the adjoint action. In practice, we use the *same* ReLN backbone across all experiments—O(2), SO(2), $D_6$ for double pendulum, Lorentz $O(1,3)$ for jet tagging, and $SO(3)/SE(3)$ with covariance for the drone task—without re-deriving group-specific kernels or solving new constraint systems. This is a key difference from EMLP-style constructions.

---

> ### Author Response · Authors · 2025-11-21
>
> ### 3. Jet tagging: comparison on a near-saturated benchmark
>
> The top-tagging benchmark is known to be nearly saturated in accuracy/AUC. To ensure a fair comparison, we:
>
> - Scaled ReLN to match the original LorentzNet parameter count (~224k);
> - Reproduced Lie Neurons [2] at the same capacity (224k).
>
> | Architecture           | #Params | Accuracy | Rej@30% ↑         |
> | :--------------------- | ------: | -------: | -----------------:|
> | LorentzNet (orig.) [5] | 224k    | 0.942    | $2195 \pm 173$    |
> | Lie Neurons [2]        | 224k    | 0.941    | $1655 \pm 73$     |
> | **ReLN (Ours)**        | 224k    | 0.942    | **$2201 \pm 161$**|
>
> At matched capacity, ReLN improves Rej@30% from $\sim 1655$ (Lie Neurons) to $\sim 2201$ (>30% relative), indicating that the reductive formulation is stronger than the semisimple Lie Neuron formulation on this benchmark and that ReLN can act as a **drop-in replacement** while retaining SOTA performance. In the revised version, we will mention extending ReLN to more complex collider tasks as a natural direction for future work.
>
> ---
>
> ### 4. Clarifications and planned revisions
>
> We plan the following clarifications and minor edits in the revised manuscript:
>
> - **Definitions (reductive algebra, Killing form, degeneracy).**
>   We will move concise definitions of reductive Lie algebras, the Killing form, and degeneracy to the Preliminaries, and explicitly state that $\mathfrak{gl}(n) \cong \mathbb{R}I \oplus \mathfrak{sl}(n)$ is reductive but not semisimple, with our $\widetilde B$ providing a non-degenerate Ad-invariant extension.
>
> - **Overview of Lie Neurons.**
>   We will add a short summary of Lie Neurons [2] in the background/related-work section and clearly note that ReLN reduces to Lie Neurons when the center is removed (our "no-center" ablations in Table 1).
>
> - **Equivariance of nonlinearity and pooling.**
>   In Eq. (4), the direction $d$ is input-dependent and obtained via an equivariant linear map from $x$; both $x$ and $d$ transform covariantly, so $B(x,d)$ is invariant and the ReLN-ReLU and Max-Killing pooling remain equivariant. We will clarify this mechanism and add a short proof in the appendix.
>
> ---
>
> In summary, the additional experiments and ablations show that:
>
> - ReLN behaves as expected on semisimple tasks,
> - **improves over both Lie Neurons and SE(3)-Transformer** when reductive structure and uncertainty are present, and
> - is competitive with or more efficient than general matrix-group methods such as EMLP on articulated dynamics,
>
> while being implemented as a **single $GL(n)$-equivariant backbone** that handles multiple subgroups without group-specific kernel engineering.
>
> ---
>
> ### References
>
> [1] Finzi et al. A Practical Method for Constructing Equivariant Multilayer Perceptrons for Arbitrary Matrix Groups. ICML 2021.
>
> [2] Lin et al. Lie Neurons: Adjoint-Equivariant Neural Networks for Semisimple Lie Algebras. ICML 2024.
>
> [3] Thomas et al. Tensor Field Networks: Rotation- and Translation-Equivariant Neural Networks for 3D Point Clouds. arXiv 2018.
>
> [4] Fuchs et al. SE(3)-Transformers: 3D Roto-Translation Equivariant Attention Networks. NeurIPS 2020.
>
> [5] Gong et al. An Efficient Lorentz Equivariant Graph Neural Network for Jet Tagging. JHEP 2022.

---

### Official Review · Reviewer_X174 · 2025-11-01

**Soundness:** 3
**Presentation:** 3
**Contribution:** 2
**Rating:** 2
**Confidence:** 3

**Summary:**

This paper proposes a new class of equivariant neural networks with inputs and outputs residing in the Lie algebra of GL(n) that are equivariant to the adjoint action of the group. Such adjoint equivariance is useful for problems involving matrix-valued data such as inertia and covariance, and can be extended to the usual left-action equivariance in some cases. While the basic formulation of adjoint-equivariant linear layers and nonlinearities has been established in a prior work [1], the method therein requires non-degeneracy of the Killing form on the Lie algebra and the contribution of this work is extending applicability to GL(n) and subgroups with reductive Lie algebra that may have degenerate Killing form. The key idea for this is leveraging the structure of reductive Lie algebras to augment the Killing form with a choice of Ad-invariant inner product on the center to resolve the nondegeneracy. With this, the authors generalize the designs of linear layer, nonlinearities, and pooling layers in [1]. The authors evaluate the proposed architecture on SO(3) adjoint equivariant Platonic solid classification task, a Sp(4) adjoint invariant synthetic regression task, SO(1, 3) left equivariant jet tagging task based on a new identification of embedding map that equates left action on data space and adjoint action on embedding space, and SO(3) adjoint equivariant drone state estimation with uncertainty represented by covariance matrix, and report improved performance mainly over non-equivariant baselines, vector neurons and Lie neurons.

[1] Lin et al., Lie Neurons: Adjoint-Equivariant Neural Networks for Semisimple Lie Algebras, ICML 2024.

**Strengths:**

S1. The paper's use of augmented bilinear form to generalize Lie neurons to reductive Lie algebras is original and technically sound as far as I can confirm.

S2. The writing and presentation is overall clear and easy to follow.

**Weaknesses:**

W1. Among the four experiments presented, as far as I can understand there are no experiments that involves reductive Lie algebras with degenerate Killing form, which are the key targets of the construction given in the paper (please correct me if I am wrong). From this, I was not able to draw the conclusion that the proposed method actually solves the problem setup given in Section 4.1.

W2. While the work extends the applicability of [1] to reductive Lie algebras, the extension might not be as broad as one may expect. One reason is that the construction cannot be applied to non-unimodular Lie groups [2, Corollary 8.31] including affine transformations (Line 39). Experiments and/or discussions on which specific groups and problems are made accessible based on this work, and why they are useful, seem necessary.

W3. I am not sure why being able to process matrix-valued inputs (e.g., in geometric uncertainty processing in Section 5.3) is a unique advantage or contribution of this work, given that standard left-action equivariant architectures offer extensions to matrix- or tensor-inputs based on actions on product spaces [3, 4, 5]. The authors discuss these approaches in Lines 109-111 but I am not convinced that these types of architectural designs are substantially more limited than the proposed method. One specific reason is that, while extending these architectures to matrix-valued inputs is direct, extending the proposed adjoint equivariant architecture to left-action problems is not as straightforward and requires problem-specific handling (e.g., Section 5.2).

[2] Knapp, Lie Groups: Beyond an Introduction, 2002.

[3] Thomas et al., Tensor field networks: Rotation- and translation-equivariant neural networks for 3D point clouds, arXiv 2018.

[4] Fuchs et al., SE(3)-Transformers: 3D Roto-Translation Equivariant Attention Networks, NeurIPS 2020.

[5] Finzi et al., A Practical Method for Constructing Equivariant Multilayer Perceptrons for Arbitrary Matrix Groups, arXiv 2021.

**Questions:**

I have no particular questions but would like to hear the authors' response on the weaknesses.

---

> ### Author Response · Authors · 2025-11-21
>
> We thank the reviewer for the thoughtful and detailed comments, and for the opportunity to clarify both the scope of our construction and the role of the new experiments. Our responses are below.
>
> ---
>
> ### Response to W1–W3: Scope of the construction and target applications
>
> #### W1 – Experiments with degenerate Killing form
>
> We agree that evaluating in the **reductive, non-semisimple regime** is crucial. Our covariance experiments in Section 5.3 are precisely in this setting.
>
> In the drone state-estimation task we work with (log-)covariance matrices in $\mathbb{R}^{3\times 3}$, modeled in $\mathfrak{gl}(3)$, which is reductive but not semisimple:
> $$
> \mathfrak{gl}(3) \cong \mathbb{R} I \oplus \mathfrak{sl}(3),
> $$
> and whose Killing form is degenerate along the central (trace) direction. The bilinear form used in our layers,
> $$
> \widetilde B(X,Y) = 2n\,\mathrm{Tr}(XY) - \mathrm{Tr}(X)\,\mathrm{Tr}(Y),
> $$
> is exactly the **non-degenerate extension** we derive in Section 4.1: it coincides with the Killing form on $\mathfrak{sl}(3)$ and induces a non-degenerate metric on the center $\mathbb{R} I$. Thus the drone covariance experiment directly targets the regime where the Killing form alone fails.
>
> To make this explicit, we added ablations that isolate the contribution of the reductive structure (center vs. semisimple parts) and compare against strong $SE(3)$-equivariant baselines.
>
> **Drone state estimation with covariance in $\mathfrak{gl}(3)$.**
>
> | Model               | Input type                | ATE ↓  | ATE% ↓ | RPE ↓  |
> |---------------------|---------------------------|-------:|-------:|-------:|
> | ResNet              | Vel + Cov     | 205.11 | 94.94  | 106.07 |
> | VN                  | Vel only             | 17.36  | 7.52   | 13.51  |
> | VN                  | Vel + Cov     | 191.78 | 88.66  | 98.39  |
> | SE(3)-Transformer   | Vel only             | 24.59  | 10.95  | 18.23  |
> | SE(3)-Transformer   | Vel + Cov     | 16.45  | 7.09   | 12.85  |
> | SE(3)-Transformer   | Vel + log-Cov | 16.83  | 7.56   | 13.34  |
> | ReLN                | Vel only             | 16.85  | 7.31   | 12.70  |
> | ReLN (full)         | Vel + Cov     | 16.49  | 7.21   | 13.02  |
> | **ReLN (full)**     | **Vel + log-Cov** | **13.92** | **5.99** | **11.04** |
> | ReLN (no-center)\*  | Vel + Cov     | 16.86  | 7.43   | 13.65  |
> | ReLN (no-center)\*  | Vel + log-Cov | 15.65  | 6.76   | 12.04  |
>
> \* “No-center” restricts to $\mathfrak{sl}(3)$ and ignores the trace in $\widetilde B$, i.e., a semisimple Lie Neurons setting.
>
> Two points address W1 directly:
>
> - **ReLN (full, velocity + log-covariance in $\mathfrak{gl}(3)$)** achieves ATE 13.92 / ATE% 5.99 / RPE 11.04, outperforming both VN/LN baselines and SE(3)-Transformers (best ATE $\approx 16.45$ with covariance features).
> - **No-center variants**, which are mathematically equivalent to the **pure semisimple Lie Neurons setting** (restricting to $\mathfrak{sl}(3)$ and ignoring the trace direction), consistently underperform the full reductive ReLN.
>
> This shows that (1) we **do** evaluate on a reductive Lie algebra with degenerate Killing form ($\mathfrak{gl}(3)$), and (2) explicitly modeling the center via our non-degenerate extension $\widetilde B$ yields clear gains over both semisimple-only variants and strong $E(3)$-equivariant baselines. We will clarify this connection to the Section 4.1 construction in the main text.
>
> ---
>
> #### W2 – On non-unimodular groups and practical scope
>
> We appreciate the reviewer’s remark on **non-unimodular Lie groups** [2]. Our construction is formulated at the level of **reductive matrix Lie algebras** (e.g., $\mathfrak{gl}(n)$, $\mathfrak{so}(p,q)$, $\mathfrak{u}(p,q)$ and finite products), which correspond to **unimodular** matrix groups. We do **not** claim applicability to arbitrary non-unimodular groups such as general affine $ax + b$ transformations, and we will state this explicitly in Section 4.1.
>
> The primary application domains of interest in this work—rigid-body geometry, uncertainty-aware state estimation, and quantum-/unitary-style constraints—are governed by unimodular groups such as
> $$
> SO(3),\quad SE(3),\quad GL(n),\quad O(p,q),\quad U(p,q),
> $$
> and their Lie algebras. These are the symmetries that appear in our experiments:
>
> - **Articulated dynamics (double pendulum):** compact / orthogonal symmetries (O(2), SO(2), $D_6$), directly comparable to EMLP [1].
> - **Top tagging:** Lorentz-equivariant jet tagging and Lie-algebraic baselines.
> - **Drone + uncertainty-aware robotics:** adjoint actions on $\mathfrak{gl}(3)$ for log-covariance matrices.
>
> We therefore position our contribution as a **broad extension within the practically relevant class of reductive matrix groups**, which already subsumes the symmetry classes most commonly used in geometric deep learning, rather than as a universal solution for all non-unimodular Lie groups.

---

> ### Author Response · Authors · 2025-11-21
>
> #### W3 – Why matrix-valued inputs are not “just” product-space features
> We agree that left-action equivariant architectures can be extended to matrix- or tensor-valued inputs by acting on product spaces [1, 3, 4], and we do not claim they cannot process such inputs. Our point is structural: in many of our target applications, matrices have a **specific algebraic meaning and transformation rule** (e.g., covariances, inertial tensors, linearized dynamics, unitary blocks). Treating them as generic product-space features does not exploit this structure.
> For example, covariances undergo congruence transformations, which in our formulation correspond to adjoint actions on $\mathfrak{gl}(n)$, and in TFN-style architectures are modeled as rank-2 (type-2) tensor representations. ReLN is built directly on the adjoint representation and the reductive decomposition
> $$
> \mathfrak{g} = z(\mathfrak{g}) \oplus [\mathfrak{g},\mathfrak{g}],
> $$
> yielding two concrete advantages:
>
> 1. **Geometry-aware covariance processing.**
> In the drone experiment, log-covariances live on an SPD manifold with congruence action. Feeding (log-)covariances as additional tensor features to SE(3)-Transformers yields only modest improvement, whereas ReLN’s adjoint action on $\mathfrak{gl}(3)$ and non-degenerate form $\widetilde B$ produces a substantial gain (ATE 13.92 vs. $\approx 16.5$ for the best SE(3)-Transformer variant).
>
> 2. **Unified “shape + scale/uncertainty”.**
>    The reductive split lets ReLN jointly model rotations (semisimple part) and scale/uncertainty (center), effectively unifying Vector Neurons and Lie Neurons within a single algebraic framework, rather than treating these as separate feature channels.
>
> We view our work as **complementary** to existing general matrix-group frameworks, which mostly focus on orthogonal/Lorentz examples and left-action benchmarks:
>
> - G-MACE [5] propose a general framework for equivariant networks on reductive Lie groups, but experiments focus on orthogonal and Lorentz symmetries.
> - GRepsNet [6] give a representation-based construction for arbitrary matrix groups, but current benchmarks again concentrate on $O(3)$, $O(5)$, and $O(1,3)$ with standard scalar/vector/second-order tensor data.
>
> To compare directly with such general methods, we additionally evaluated ReLN on the **double-pendulum Hamiltonian dynamics benchmark** introduced by EMLP and measured both accuracy and computational cost.
>
> **Table 2: Mean rollout error on the EMLP double-pendulum benchmark.**
> *ReLN achieves equal or lower error than the original EMLP across all symmetry groups.*
>
> | Task | EMLP O(2) | **Ours O(2)** | EMLP SO(2) | **Ours SO(2)** | EMLP $D_6$ | **Ours $D_6$** | MLP (Id) |
> |:-----|----------:|--------------:|-----------:|---------------:|-----------:|---------------:|---------:|
> | HNNs | 0.012(2)  | **0.011(2)**  | 0.015(3)   | **0.010(4)**   | 0.013(2)   | **0.011(2)**   | 0.028    |
>
> **Table 3: Computational efficiency (FLOPs) on the HNN task.**
> *ReLN is approximately $11\times$ more efficient than EMLP at comparable accuracy.*
>
> | Model        | \#Params | FLOPs / step |
> |:-------------|---------:|-------------:|
> | MLP-HNN      | 34,817   | 70,400       |
> | EMLP-HNN     | 55,569   | 1,589,909    |
> | ReLN-HNN | 69,889   | 142,190  |
>
> These experiments show that our model matches or improves EMLP on the double-pendulum benchmark across all symmetry groups (O(2), SO(2), $D_6$) while being roughly $11\times$ cheaper in FLOPs per step. This directly addresses W3: even compared to highly general matrix-group frameworks, the proposed adjoint/reductive construction is not only structurally different but also more **computationally practical**.
>
> We therefore view these works and ours as complementary responses to the same open problem: making “general equivariance” for matrix/reductive groups truly practical beyond orthogonal and Lorentz examples. Our contribution is to push this frontier specifically for **adjoint-equivariant, reductive settings on $\mathfrak{gl}(n)$** with matrix-valued quantities such as (log-)covariances, and to show that exploiting the reductive structure yields tangible gains on realistic two dynamics benchmarks over strong baselines such as SE(3)-Transformers and EMLP.
>
> ---
>
> ### References
>
> [1] Finzi et al. A Practical Method for Constructing Equivariant Multilayer Perceptrons for Arbitrary Matrix Groups. ICML 2021.
> [2] Knapp, A. W. Lie Groups Beyond an Introduction,2nd ed. Birkhäuser, 2002.
> [3] Thomas et al. Tensor Field Networks: Rotation- and Translation-Equivariant Neural Networks for 3D Point Clouds. arXiv, 2018.
> [4] Fuchs et al. SE(3)-Transformers: 3D Roto-Translation Equivariant Attention Networks. NeurIPS 2020.
> [5] Batatia et al. A General Framework for Equivariant Neural Networks on Reductive Lie Groups. NeurIPS 2023.
> [6] Basu et al. G-RepsNet: A Lightweight Construction of Equivariant Networks for Arbitrary Matrix Groups. TMLR 2025

---

### Official Review · Reviewer_V7Ta · 2025-11-01

**Soundness:** 4
**Presentation:** 3
**Contribution:** 2
**Rating:** 4
**Confidence:** 3

**Summary:**

This paper introduces a modification to Lie Neurons (Lin et al 2024a) to make them work for reductive Lie groups. Lie Neurons introduced a clever idea for constructing equivariant nonlinearities and pooling by using the inner product define on the Lie algebra, called the Killing form. Lie neurons worked well for "semisimple" Lie groups, which don't have nontrivial center (subgroup commuting with the group), but did not work for reductive groups )having nontrivial center) because the Killing form becomes degenerate (zero norm for center). The general linear group $GL(n) \\simeq SL(n) \\rtimes \\mathbb{R}\\backslash \\{0\\} $   is reductive and its Lie algebra is $\\mathfrak{gl}(n) = \\mathbb{R} I \oplus \\mathfrak{sl}(n)$. They propose using a modified Killing form $\tilde{B}$ which adds a nondegenerate inner product in the center of the group to the usual Killing form.
They use $\tilde{B}$ instead of $B$ to define nonlineariries and other layer components similar to Lie Neurons. They vallidate their layer on a few problems and show strong improvement in one, the drone navigation problem.

**Strengths:**

1. The theoretical foundation seems solid. It resolved the Killing form degeneracy issue in a very simple way.
2. ReLN usifies previous approaches as it is a slight generalization of Lie Neurons and reduces to it on semisimple Lie algebras.
3. The drone navigation results show impressive improvement over the baseline.

**Weaknesses:**

1. My main issue is contribution. Beyond defining $\tilde{B}$, I don't seem to find any distinction between this work and Lie Neurons (Lin 2024a). Specifically, the Killing form on $\\mathfrak{gl}(n)$  is $B(X,Y) = 2n\\cdot \\mathrm{tr} (XY) - 2\\mathrm{tr}(X) \\mathrm{Y}$. They define $\\tilde{B}(X,Y) = \\mathrm{tr}(X) \\mathrm{Y} + B(X,Y)$. That's basically their main theoretical contribution. All the rest, including ReLN-ReLU nonlinearity and ReLN-Bracket seem to be identical to Lie Neurons, just using $\tilde{B}$ instead of $B$.
2. For the drone experiment, the VN baseline seems a bit strange. As vector neurons don't work for matrices and for the velocity+covariance case they do eigendecomposition. But compared to velocity alone, the results with covariance are an order of magnitude worse. So, the eigendecompositon proposal may be flawed. As I understand it, no one else has suggested that VN can be used this way, right? So maybe a naive baseline like SO(3) augmented ResNet would have been more competitive. In contrast, the velocity only results onf VN are not bad at all and pretty close to ReLN.

**Questions:**

1. can you clarify the distinction between this work and Lin 2024a beyond $\tilde{B}$? some of the contribution points seem too specific and don't give a big picture, beyond a slightly different application of Lie Neurons.
2. In the top-tagging, the results are basically the same as some baselines. Is there an advantage in using ReLN? Could Lie Neuron be applied there too?
3. did you consider stronger baselines for the drones? Could maybe combining using VN for the velocity with MLP for covariance help?

---

> ### Author Response · Authors · 2025-11-21
>
> We thank the reviewer for the insightful comments. We appreciate the opportunity to clarify how our work fundamentally extends Lie Neurons and how it behaves against stronger baselines.
>
> ---
>
> ## 1. GL(n) equivariance: a single framework for reductive geometries and subgroups
> **(Addressing W1, Q1)**
>
> The reviewer asked whether ReLN is only a moderate extension of Lie Neurons. Our contribution is **not an incremental update**, but a universal adjoint-equivariant framework on \(GL(n)\) that
>
> 1. solves equivariant problems across all subgroups without task-specific architecture design, and
> 2. moves from the **semisimple regime** of Lie Neurons [2] to a **general reductive setting** with concrete algebraic and modeling consequences.
>
> - **All-in-one equivariance (vs. EMLP).**
>   Unlike EMLP [1], which requires specifying each symmetry group and solving analytic constraints to build equivariant bases per layer, ReLN uses a **single, efficient architecture** for any symmetry that embeds into \(GL(n)\). Exact equivariance is enforced by Lie-algebra operations (bracket + \(\widetilde B\)), without constraint solving.
>
> - **Matrix-valued, non-semisimple inputs.**
>   Many real-world objects (covariances, inertia matrices, linearized dynamics) live naturally in $\mathfrak{gl}(n)$, carrying scale and anisotropy. ReLN is, to our knowledge, first framework to process these **strictly non-semisimple inputs** directly in their native algebraic form, independent of the specific underlying symmetry.
>
> ### 1.A Beyond heuristics: the reductive derivation (W1)
>
> For \(GL(n)\), our bilinear form is
> $$
> \widetilde B(X,Y) = 2n\,\mathrm{Tr}(XY) - \mathrm{Tr}(X)\,\mathrm{Tr}(Y).
> $$
> This could look ad-hoc if viewed in isolation, but in the paper it is derived from a **general construction on reductive Lie algebras**
> $$
> \mathfrak{g} = z(\mathfrak{g}) \oplus [\mathfrak{g},\mathfrak{g}],
> $$
> by combining
>
> - the Killing form on the semisimple ideal $[\mathfrak{g},\mathfrak{g}]$, and
> - an Ad-invariant inner product on the center $z(\mathfrak{g})$,
>
> into a single non-degenerate, Ad-invariant form \(\widetilde B\).
> For $\mathfrak{gl}(n)$, this yields a **canonical trace-based instantiation**: Killing form on $\mathfrak{sl}(n)$ plus a standard inner product on the trace direction. This is precisely what guarantees mathematical stability in settings where the Killing form is degenerate (e.g., \(GL(n)\), \(U(p,q)\), products \(GL(n_1)\times GL(n_2)\)), and is what enables us to build valid adjoint-equivariant layers beyond the semisimple case.
>
> ### 1.B Unifying semisimple and abelian geometries (Q1)
>
> Vector Neurons (VN) [9], which act on \(\mathbb{R}^3\) with an \(SO(3)\) action, and Lie Neurons, which act on semisimple algebras via the Killing form, both arise as **special cases** of our reductive bilinear form. This gives a clean explanation of how ReLN:
> - **subsumes VN-style architectures** (and extensions such as VN-Transformers), and
> - **extends** them to non-compact, non-semisimple symmetries that couple rotations with scaling and shear.
> This is what allows us to treat **matrix-valued quantities** (e.g., covariances) as genuine geometric objects:
>
> - **3D Gaussian splatting & navigation.**
>   In 3DGS-based navigation [4,5], scenes are represented by anisotropic Gaussians. ReLN operates directly on **covariance ellipsoids**, enabling equivariant message passing on splat geometry instead of flattening them into vectors.
> - **Quantum & complex groups.**
> ReLN applies to general **reductive** algebras, including
> $\mathfrak{u}(n) \cong \mathfrak{su}(n) \oplus \mathfrak{u}(1),$
>   and explicitly models the central phase $\mathfrak{u}(1)$ direction that Lie Neurons discard. This makes ReLN compatible with unitary parameterizations and PEFT methods such as Quantum-PEFT [10], where Pauli-rotation–based updates yield logarithmic parameter scaling.
> - **Geometric diffusion & noise modeling.**
>   Score-based generative models rely on noise scales. The reductive structure of ReLN jointly represents **noise scale and geometry**, providing a natural way to build equivariant score-based models [6].
>
> Taken together, these points address Q1: ReLN is not just a different bilinear form on a fixed semisimple algebra, but a **broader reductive framework** that unifies semisimple and abelian components and enables applications where matrix-valued geometry, uncertainty, and unitary constraints are central.

---

> ### Author Response · Authors · 2025-11-21
>
> ## 2. Superiority over strong SE(3) and EMLP baselines
> **(Addressing W2, W3, Q3)**
> The reviewer correctly noted that “VN + eigendecomposition” is a weak baseline for matrix inputs. To strengthen the evidence, we added strong benchmarks showing that ReLN outperforms:
> - a \(SE(3)\)-equivariant baseline (SE(3)-Transformer [3]),
> - a general matrix-group method (EMLP [1]), and
> - semisimple-only Lie Neurons [2] in the reductive regime.
>
> ### 2.A Drone state estimation: outperforming SE(3)-Transformers
> We evaluate on a drone state-estimation task where covariances/log-covariances live in
> $\mathfrak{gl}(3) \cong \mathbb{R} I \oplus \mathfrak{sl}(3),$
> so the Killing form is degenerate on the center.
>
> **Table 1: Drone state estimation.**
> *Best result in bold.*
>
> | Model                | Input type        | ATE ↓  | ATE% ↓ | RPE ↓  |
> |----------------------|-------------------|-------:|-------:|-------:|
> | VN                   | Vel only          | 17.36  | 7.52   | 13.51  |
> | VN                   | Vel + Cov         | 191.78 | 88.66  | 98.39  |
> | SE(3)-T    | Vel only          | 24.59  | 10.95  | 18.23  |
> | SE(3)-T    | Vel + Cov         | 16.45  | 7.09   | 12.85  |
> | SE(3)-T    | Vel + log-Cov     | 16.83  | 7.56   | 13.34  |
> | ReLN                 | Vel only          | 16.85  | 7.31   | 12.70  |
> | ReLN (full)          | Vel + Cov         | 16.49  | 7.21   | 13.02  |
> | **ReLN (full)**      | **Vel + log-Cov** | **13.92** | **5.99** | **11.04** |
> | ReLN (no-center)\*   | Vel + Cov         | 16.86  | 7.43   | 13.65  |
> | ReLN (no-center)\*   | Vel + log-Cov     | 15.65  | 6.76   | 12.04  |
>
> \* “No-center” restricts to $\mathfrak{sl}(3)$ and ignores the trace in $\widetilde B$: a semisimple Lie Neurons setting.
>
> **Analysis.**
> ReLN (full) with velocity + log-covariance achieves the best results, clearly improving over the strongest SE(3)-T setting and over the no-center variant; **this improvement is substantial on a large-scale drone motion dataset (~200 m scale).** Simply switching SE(3)-T from covariance to log-covariance does not help, whereas ReLN benefits significantly from the same representation. This shows that explicitly modeling the degenerate killing form is crucial in this uncertainty-aware, reductive setting.
> ### 2.B Learning articulated object dynamics (EMLP benchmark)
> We also evaluate on the **double-pendulum Hamiltonian dynamics benchmark** introduced by EMLP.
>
> **Table 2: Test rollout error on EMLP double-pendulum benchmark.**
>
> | Task | EMLP O(2) | **Ours O(2)** | EMLP SO(2) | **Ours SO(2)** | EMLP \(D_6\) | **Ours \(D_6\)** | MLP |
> | :--- | --------: | ------------: | ----------: | -------------: | ------------: | ---------------: | -------: |
> | HNNs | 0.012(2)  | **0.011(2)**  | 0.015(3)    | **0.010(4)**   | 0.013(2)      | **0.011(2)**     | 0.028    |
>
> ### 2.C Computational efficiency: removing the constraint bottleneck
>
> **Table 3: Computational efficiency (FLOPs) on HNN task.**
>
> | Model        | #Params | FLOPs / step |
> | :----------- | ------: | -----------: |
> | MLP-HNN      | 34,817  | 70,400       |
> | EMLP-HNN | 55,569  | 1,589,909    |
> | ReLN-HNN | 69,889  | 142,190  |
>
> ReLN **matches or improves** EMLP’s accuracy while using roughly \(11\times\) fewer FLOPs per step, directly addressing the reviewer’s concerns about practicality and efficiency.
>
> ---
>
> ## 3. Top-tagging: robustness and consistency
> **(Addressing Q2)**
> Finally, we directly compare against Lie Neurons [2] on the top-tagging task:
> - **ReLN (ours):** Rej@30% = **2201 ± 161** (224k params)
> - **Lie Neurons:** Rej@30% = **1655 ± 73** (224k params)
> Classification accuracy is saturated across methods, but ReLN achieves a significantly better background rejection rate at matched parameter count. This shows that our generalization from Lie Neurons to ReLN **does not incur a performance penalty** on semisimple-dominated tasks, while enabling the broader reductive framework described above.
>
> ---
>
> ### References
>
> [1] Finzi et al. A Practical Method for Constructing Equivariant Multilayer Perceptrons for Arbitrary Matrix Groups. ICML 2021.
> [2] Lin et al. Lie Neurons: Adjoint-Equivariant Neural Networks for Semisimple Lie Algebras. ICML 2024.
> [3] Fuchs et al. SE(3)-Transformers: 3D Roto-Translation Equivariant Attention Networks. NeurIPS 2020.
> [4] Kerbl et al. 3D Gaussian Splatting for Real-Time Radiance Field Rendering. ACM Trans. Graph. (TOG) 2023.
> [5] Matsuki et al. Gaussian Splatting SLAM. CVPR 2024.
> [6] Hoogeboom et al. Equivariant Diffusion for Molecule Generation in 3D. ICML 2022.
> [7] Thomas et al. Tensor Field Networks: Rotation- and Translation-Equivariant Neural Networks for 3D Point Clouds. arXiv, 2018.
> [8] Gong et al. An Efficient Lorentz Equivariant Graph Neural Network for Jet Tagging. JHEP 2022.
> [9] Denget al. Vector Neurons: A General Framework for SO(3)-Equivariant Networks. ICCV 2021.
> [10] Koike-Akino, et al. Quantum-PEFT: Ultra Parameter-Efficient Fine-Tuning. ICLR 2025.

---

### Official Review · Reviewer_ChwN · 2025-11-02

**Soundness:** 2
**Presentation:** 3
**Contribution:** 2
**Rating:** 4
**Confidence:** 4

**Summary:**

The paper ‘Equivariant Neural Networks for General Linear Symmetries on Lie Algebras’ suggests an approach how to adopt Lie Neurons architecture (Lin et al., 2024) to the case of the tasks when the Killing form is degenerate (this happens if the corresponding Lie algebra is reductive). This problem of Lie Neurons is mentioned in the discussion section of their paper. The authors of the current paper call their modification of Lie Neurons architecture as Reductive Lie Neurons (ReLNs). ReLNs are equivariant with respect to the adjoint action of the general linear Lie group and, consequently, all its subgroups. ReLNs propose a learnable, non-degenerate, and Ad-invariant bilinear form that is effective for the reductive (but non-semisimple) algebra gl(n). The authors demonstrate the framework's versatility on algebraic benchmarks (semisimple case), a Lorentz-equivariant particle physics task (via an embedding into a reductive algebra), and a 3D drone state estimation problem (via an embedding into another reductive algebra).

**Strengths:**

1). The paper is generally well-written. I liked the introduction and Figures 1 and 2. They provide nice high-level overviews of the application landscape and the method's position in the field.
\
2). Providing a practical architecture for exact GL(n) equivariance is an original contribution (although direct applications of full GL(n)-equivariance are questionable).

**Weaknesses:**

1). Theoretical and conceptual contribution of the work is moderate:
\
a). When the Lie algebra is semisimple, the proposed form $\tilde{B}$ almost coincides with the Killing form $B$ considered in Lie Neurons (see Section C.1). But the semisimple Lie algebras are mostly interesting for applications, and even the authors' experiments in Sections 5.1.1 and 5.1.2 consider the tasks with semisimple Lie algebras, and in the experiment in Section 5.2, the authors artificially put the O(1,3)-equivariant task in the reductive gl(5).
\
b). The proposed layers (Section 4.2) mostly duplicate the layers proposed in Lie Neurons, just changing $B$ on $\tilde{B}$.
\
c). The proofs of the main theorems (Proposition 4.1 and equivariance of the proposed mappings; the proofs are in Propositions B.1-B.3 and Appendix D)  seem to appear either almost trivial or almost line by line similar to those provided in the Lie Neurons paper.

2). No public code is provided, which hinders verification of the reported results and adoption of the method.

3). The current experiments 5.1.1 and 5.1.2 duplicate the ones provided in Lie Neurons (for semisimple Lie algebras). However, I believe the experiments are insufficient for the provided statements. They do not fully substantiate the need for the generalized gl(n) framework. Key benchmarks (Sections 5.1.1, 5.1.2) are performed on semisimple algebras where ReLNs and Lie Neurons are nearly equivalent. In Section 5.2, the Lorentz-equivariant task is (probably auxiliary) embedded into gl(5), but a comparison against Lie Neurons applied to the native semisimple algebra is missing, making it difficult to judge the necessity of the reductive extension.

4). The authors mention only SO(n) equivariance in introduction and related work, however O(n), O(p,q), SO(p,q) are also often applied (the authors even use them in the top tagging experiment).

5). The related work section omits a significant body of literature on equivariant networks based on Clifford (Geometric) algebras (see, for example, Clifford Group Equivariant Neural networks (CGENN, https://openreview.net/forum?id=n84bzMrGUD&noteId=sQG6abJbs8), Geometric Algebra Transformer (https://openreview.net/forum?id=M7r2CO4tJC), Clifford Group Equivariant Simplicial Message Passing Networks (https://openreview.net/forum?id=Zz594UBNOH&noteId=6jug7zZnrW), Generalized Lipschitz Groups Equivariant Neural Networks (https://openreview.net/forum?id=H0ySAzwu8k), and others). The paper CGENN can be added in Top Tagging experiment 5.2, they also consider this task.

6). Minor points:
\
a). Consistency of notation: the authors use both gl(n) and gl(n,R) notion for the same Lie algebra (e.g. see lines 159 and 212).
\
b). In the references, all the titles of the papers are lowercased, so sometimes ‘Lie’ appears as ‘lie’, etc.
\
c). Formula (17) duplicates the proof of Proposition A.1.

**Questions:**

1). What are the results for the original Lie Neurons model on the Top Tagging (5.2) and Drone State Estimation (5.3) tasks? A direct comparison is crucial for evaluating the improvement offered by the ReLN generalization of Lie Neurons.

2). I did not fully understand the following point. The paper proves equivariance under the adjoint action ($X\mapsto gXg^{-1}$) of the general linear group GL(n) and its subgroups. For orthogonal groups, the standard notion of equivariance is often defined by the left multiplication for vectors ($v\mapsto gv$), i.e. we can either firstly multiply any vector by an orthogonal matrix and then apply the network, or firstly apply the network and then multiply the result by the same orthogonal matrix, and we need to get the same result. Could you please clarify the relationship between these two actions in your framework, especially for groups like O(n) and SO(n)? Does the group adjoint action become a left multiplication on the R^3 for the orthogonal groups? Please provide the details.

3). The Killing form for gl(n) has the form $B(x,y)=2n \cdot trace(XY)-2\cdot trace(X)\cdot trace(Y)$ (see e.g. Wiki page on the Killing form). The authors introduce the modified Killing form $\tilde{B}=2n \cdot trace(XY)-trace(X)\cdot trace(Y)$ (without 2 in the second summand). Could you please specify how these 2 formulas are related? What was the motivation for the specific scaling in your definition? Any details would be possibly helpful for the readers.

4). In Table 3, what is the precise meaning of "Test Aug." and the reported values? Is this MSE?

5). For the reader's convenience, could you add a brief table in the appendix summarizing the definitions of the main Lie groups and algebras considered in the work besides general linear (special linear, symplectic, etc.)?

6). Can the ReLN architecture be generalized to complex Lie groups and algebras, which are also prevalent in applications like quantum mechanics?

7). Please address comments, suggestions, and questions in my Weaknesses section.

---

> ### Author Response · Authors · 2025-11-21
>
> We thank the reviewer for the thorough assessment and constructive feedback. We appreciate the recognition of our method’s mathematical rigor. Below, we address the concerns regarding theoretical novelty and experimental sufficiency. We will incorporate these changes into a revised version of the manuscript.
>
> ---
>
> ## 1. GL(n) Equivariance: A Single Framework for All Subgroups
> The reviewer expressed concern that ReLN might be a moderate extension of Lie Neurons. We respectfully emphasize that our contribution is **not an incremental update, but a universal framework for $GL(n)$ that solves equivariant problems across all subgroups without task-specific architecture design.**
>
> - **All-in-one Equivariance (vs. EMLP).** Unlike EMLP [1], which requires specifying exact symmetry groups and solving expensive constraints for each layer's basis, ReLN provides a **single, efficient architecture** capable of handling any problem embedded in $GL(n)$. This eliminates the computational bottleneck of analytic constraint solving while maintaining exact equivariance.
> - **Applicability to General Matrix Inputs.** Whether the task involves **point clouds** (e.g., [7]), **graph message passing** (e.g., [8, 9]), or **dynamics** (articulated objects; [1]), real-world data  manifests as general matrices ($\mathfrak{gl}(n)$) carrying scale and anisotropy. ReLN is, to our knowledge, the first framework to process these **strictly non-semisimple inputs** directly in their native algebraic form, independent of the specific underlying symmetry. This unification allows ReLN to process **mixed features** (e.g., velocities + covariance) as unified geometric objects.
>
> ---
>
> ## 2. Experimental validation on complex dynamics
>
> We added **two benchmarks** showing that ReLN outperforms strong $SE(3)$-equivariant baselines (SE(3)-Transformers [3]), general matrix-group solvers (EMLP [1]), and semisimple-only Lie Neurons [2] in the reductive regime.
>
> ### A. Drone State Estimation with Strong SE(3) Baseline
>
> We updated the Drone State Estimation task to include the SE(3)-Transformer and Lie Neurons.
>
> **Table 1: Performance on the Drone State Estimation Dataset.**
> *Best result in bold.*
>
> | Model                | Input type                    | ATE ↓  | ATE% ↓ | RPE ↓  |
> |----------------------|--------------------------------|-------:|-------:|-------:|
> | SE(3)-Transformer    | Vel only                 | 24.59  | 10.95  | 18.23  |
> | SE(3)-Transformer    | Vel + Cov         | 16.45  | 7.09   | 12.85  |
> | SE(3)-Transformer    | Vel + log-Cov     | 16.83  | 7.56   | 13.34  |
> | ReLN                 | Vel                 | 16.85  | 7.31   | 12.70  |
> | ReLN (full)          | Vel + Cov         | 16.49  | 7.21   | 13.02  |
> | **ReLN (full)**      | **Vel + log-Cov** | **13.92** | **5.99** | **11.04** |
> | ReLN (no-center)*    | Vel + Cov         | 16.86  | 7.43   | 13.65  |
> | ReLN (no-center)*    | Vel + log-Cov     | 15.65  | 6.76   | 12.04  |
>
> \* The “no-center” variants are mathematically equivalent to the semisimple Lie Neurons setting: the Killing form is restricted to $\mathfrak{sl}(3)$ and the central trace direction is ignored in $\widetilde B$.
>
> **Key observations.**
> ReLN (full) clearly outperforms both SE(3)-Transformer and the no-center variant; this improvement is substantial on a large-scale drone motion dataset (~200 m scale). Switching SE(3)-Transformer from covariance to log-covariance does not help, whereas ReLN benefits strongly from the same representation. Modeling covariances in the **full** reductive algebra, rather than in the semisimple part alone. These results both strengthen our baseline comparison (vs. SE(3)-Transformer and Lie Neurons) and directly support our claim that ReLN’s reductive construction is effective exactly in the degenerate-Killing-form regime.
>
> ### B. Articulated Object Dynamics (EMLP Benchmark)
>
> We also evaluate on the **double-pendulum Hamiltonian dynamics benchmark** introduced by EMLP.
>
> **Table 2: Test rollout error on EMLP Double-Pendulum Benchmark.**
> *ReLN achieves lower error than the original EMLP across all symmetry groups.*
>
> | Task | EMLP O(2) | **Ours O(2)** | EMLP SO(2) | **Ours SO(2)** | EMLP $D_6$ | **Ours $D_6$** | MLP |
> | :--- | --------: | ------------: | ----------: | -------------: | ----------: | -------------: | -------: |
> | HNNs | 0.012(2)  | **0.011(2)**  | 0.015(3)    | **0.010(4)**   | 0.013(2)    | **0.011(2)**   | 0.028    |
>
> ### C. Computational Efficiency vs. EMLP
>
> **Table 3: Computational Efficiency (FLOPs) on HNN Task.**
> *ReLN is approximately $11\times$ more efficient than EMLP.*
>
> | Model      | #Params | FLOPs / step |
> | :--------- | ------: | -----------: |
> | MLP-HNN    | 34,817  | 70,400       |
> | EMLP-HNN [1] | 55,569 | 1,589,909    |
> | ReLN-HNN | 69,889 | 142,190 |
>
> **Conclusion.**
> As shown in Table 3, ReLN offers a **scalable, efficient alternative** that maintains exact equivariance without the computational bottleneck, making it more suitable for larger datasets.

---

> ### Author Response · Authors · 2025-11-21
>
> ## 3. Top-Tagging: Robustness and Consistency
> **(Addressing Q1)**
>
> We reproduced the original Lie Neurons on the top-tagging task for a direct comparison:
>
> - **ReLN (ours):** Rej@30% = **2201** $\pm$ 161 (224k params)
> - **Lie Neurons:** Rej@30% = **1655** $\pm$ 73 (224k params)
> While accuracy is saturated, ReLN achieves the performance ceiling on the stricter rejection metric. This confirms that ReLN serves as a robust **“drop-in” generalization** that entails no performance penalty.
>
> ---
>
> ## 4. Theoretical Clarifications & Extensions
> **(Addressing Q2, Q3, Q6)**
>
> - **Adjoint vs. left multiplication (Q2).**
>     For general $n$, adjoint equivariance on $\mathfrak{so}(n)$ is distinct from left multiplication on $\mathbb{R}^n$. Only for $n = 3$  we have a representation isomorphism $\mathbb{R}^3 \cong \mathfrak{so}(3)$, which makes the adjoint action equivalent (via the hat/vee map) to the familiar vector action. This explains how ReLN unifies “vector” and “Lie” behaviors in the SO(3) case, but not in higher dimensions.
>
> - **Bilinear form scaling (Q3).**
>   The difference in coefficients from the classical Killing form is not arbitrary; it is a direct consequence of defining a non-degenerate metric on the center of the algebra. Our modified coefficient is chosen so that $\widetilde B$ coincides with the Killing form on the semisimple part and matches a natural inner product on the central $\mathbb{R}$ direction. In general there is no canonical choice of this bilinear form on the center; once non-degeneracy is satisfied, the specific coefficient simply determines the relative metric scaling between semisimple components and the central (scaling) component.
>
> - **Quantum and complex groups (Q6).**
>   ReLN applies to general **reductive** Lie algebras, so it naturally covers **unitary settings** such as
>   $\mathfrak{u}(n) \cong \mathfrak{su}(n) \oplus \mathfrak{u}(1)$,
>   including the central phase ($\mathfrak{u}(1)$) direction that Lie Neurons ignore. This makes ReLN applicable to quantum-inspired parameterizations and parameter-efficient fine tuning methods that leverage unitary weight updates, such as Quantum-PEFT [4]. In this sense, ReLN provides a Lie-algebraic backbone for future quantization-style adapters in large-scale vision models and LLMs.
>
> ---
>
> ## 5. Reproducibility & Minor Points
> **(Addressing W2, W4, W5, W6)**
>
> - **Code availability (W2).** We provide an anonymized repository:
>   <https://github.com/anon617/reln-anon>
> - **Groups beyond SO(n) (W4).** We now explicitly mention and discuss O(n), O(p,q), and SO(p,q), and clarify how they appear in the top-tagging and double-pendulum experiments.
> - **Related work (W5).** We will include citations for CGENN [5] and Geometric Algebra Transformers [6].
> - **Definitions and “Test Aug” (Q4, Q5).** We added a summary table of Lie groups in Appendix A (GL, SL, O(p,q), Sp, etc.). “Test Aug” refers to test-time augmentation; the reported values are MSE .
> - **Minor corrections (W6).** We standardized notation to $\mathfrak{gl}(n)$ and corrected bibliography capitalization (e.g., “Lie” vs. “lie”).
>
> ---
>
> ### References
>
> [1] Finzi et al. A Practical Method for Constructing Equivariant Multilayer Perceptrons for Arbitrary Matrix Groups. ICML 2021.
> [2] Lin et al. Lie Neurons: Adjoint-Equivariant Neural Networks for Semisimple Lie Algebras. ICML 2024.
> [3] Fuchs et al. SE(3)-Transformers: 3D Roto-Translation Equivariant Attention Networks. NeurIPS 2020.
> [4] Koike-Akino et al. Quantum-PEFT: Ultra Parameter-Efficient Fine-Tuning. ICLR 2025.
> [5] Ruhe et al. Clifford Group Equivariant Neural Networks. NeurIPS 2023.
> [6] Brehmer et al. Geometric Algebra Transformers. NeurIPS 2023.
> [7] Thomas et al. Tensor Field Networks: Rotation- and Translation-Equivariant Neural Networks for 3D Point Clouds. arXiv 2018.
> [8] Gong et al. An Efficient Lorentz Equivariant Graph Neural Network for Jet Tagging. JHEP 2022.
> [9] Deng et al. Vector Neurons: A General Framework for SO(3)-Equivariant Networks. ICCV 2021.

---

### Note · Program_Chairs · 2025-11-25
**Submission Desk Rejected by Program Chairs**

Desk rejected because the following twitter post breaks anonymity:
https://x.com/GhaffariMaani/status/1991905540163743927?t=-NtVbY5FXighVy0r9afsDQ&s=19